# The Lie We Tell: Correcting the Euclidean Fallacy in Vision Language Action Policies via Score Matching on Tangent Space

Bing-Cheng Chuang [* 1]   I-Hsuan Chu [* 1]   Bor-Jiun Lin [1]   YuanFu Yang [2]   Min Sun [3]   Chun-Yi Lee [1]

## Abstract

Diffusion-based Vision-Language-Action policies achieve remarkable success in robotic manipulation, yet commit a fundamental geometric error we term the **Euclidean Fallacy**: representing SE(3) poses as flat $\mathbb{R}^{12}$ vectors. This approximation induces (1) manifold drift violating SO(3) constraints, (2) broken equivariance under coordinate transformations, and (3) non-geodesic trajectories with excessive kinematic cost. We introduce **Lie Diffuser Actor (LDA)**, a diffusion framework operating intrinsically on SE(3). Our method injects noise through left-invariant SDEs, predicts scores in the tangent space, and retracts samples via the exponential map. This formulation eliminates manifold drift by construction while guaranteeing coordinate-frame equivariance and geodesic optimality. On CALVIN ABC→D, LDA improves average task length from 3.27 to 3.51 (+7.3%). We further validate our method on real robot and the results show that our methodology outperforms the baseline on majority tasks.

## 1. Introduction

Vision-Language-Action (VLA) policies have transformed robotic manipulation, enabling robots to execute complex tasks through instructions and visual observations. Central to this progress is pose trajectory generation: specifying how a robot gripper should position and orient itself over time. Early VLM-based policies (Kim et al., 2024; Qu et al., 2025; Liu et al., 2025; Zhang et al., 2025; Cen et al., 2025; Brohan et al., 2022; 2023) formulated control as token generation,

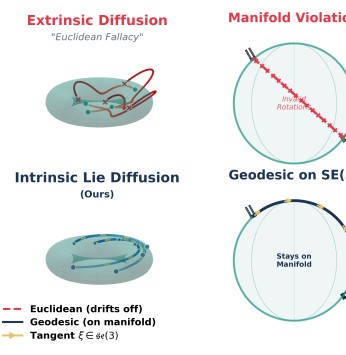

*Figure 1.* **Euclidean Diffusion vs. Lie Diffusion.** (Top) Extrinsic diffusion treats the curved SE(3) manifold as flat $\mathbb{R}^n$, which causes drift and produces invalid geometries. (Bottom) The proposed Lie Diffuser Actor injects noise as tangent twists $\boldsymbol{\xi} \in \mathfrak{se}(3)$, ensuring that trajectories follow valid geodesics.

but discretization introduces quantization errors and loses geometric structure essential for precise manipulation. Hierarchical approaches (Bjorck et al., 2025; Li et al., 2025; Cui et al., 2025; Black et al., 2024; 2025; Cheang et al., 2025; Chen et al., 2025a; Wen et al., 2025; Huang et al., 2025) separate semantic reasoning from trajectory generation, yet geometric inconsistencies persist in low-level controllers. Trajectory-level diffusion policies (Liu et al., 2024; Cheng et al., 2024; Ze et al., 2024; Ke et al., 2024) represent the current state of the art, treating entire action sequences as samples from a generative model. Despite capturing multimodal behaviors and achieving superior long-horizon consistency, a fundamental bottleneck remains: these methods parameterize SE(3) poses as flat Euclidean vectors and apply additive Gaussian noise, violating the Lie group structure and forfeiting equivariance to rigid transformations.

This geometric mismatch constitutes what we term the **Euclidean Fallacy**: the structural incompatibility between Euclidean noise injection and the SE(3) manifold, rather than a criticism of the engineering quality of prior diffusion-based policies. The Special Euclidean group SE(3) is a curved Riemannian manifold, not a flat vector space, and the operations that are valid in $\mathbb{R}^n$ do not preserve the underlying rigid transformation structure. As illustrated in Fig. 1 (top), linear interpolation departs from the valid manifold and traverses physically unrealizable configurations such as nonorthogonal rotation matrices. Three failure modes emerge from this approximation. First, *manifold drift*: generated poses violate SO(3) constraints and force networks to learn correction projections rather than manipulation semantics. Second, *broken equivariance*: Euclidean noise distributions

---

*Equal contribution   [1]Department of Computer Science and Information Engineering & Artificial Intelligence Center of Research Excellence, National Taiwan University, Taipei, Taiwan [2]Institute of Artificial Intelligence Innovation, National Yang Ming Chiao Tung University, Hsinchu, Taiwan [3]Department of Electrical Engineering, National Tsing Hua University, Hsinchu, Taiwan. Correspondence to: Chun-Yi Lee < cylee@csie.ntu.edu.tw >.

*Proceedings of the 43rd International Conference on Machine Learning*, Seoul, South Korea. PMLR 306, 2026. Copyright 2026 by the author(s).

do not transform covariantly under rigid transformations of the workspace. As a result, the learned score function becomes coordinate-frame dependent rather than intrinsic to the manipulation task. Third, *suboptimal trajectories*: Euclidean interpolation fails to capture screw motions, the kinematically natural helical paths of Chasles' theorem.

We introduce **Lie Diffuser Actor (LDA)**, a diffusion framework operating intrinsically on SE(3). Our method defines forward noising through left-invariant SDEs, injecting noise as velocity twists $\xi \in \mathfrak{se}(3)$ and mapping samples onto the manifold via the exponential map. This construction provides three theoretical guarantees. First, left-invariant noise injection eliminates manifold drift by construction (**Proposition** 4.1), as the exponential map guarantees $\exp(\xi) \in SE(3)$ for any $\xi \in \mathfrak{se}(3)$. Second, the resulting policy satisfies left-invariant equivariance (**Theorem** 4.2): transforming the workspace by $h \in SE(3)$ produces equivalently transformed outputs without retraining. Third, reverse-time dynamics generate Riemannian geodesics (**Proposition** 4.3), yielding kinematically optimal screw motions. Full proofs appear in Appendix A.4.1.

Empirical results validate these theoretical properties. On CALVIN ABC→D, Lie Diffuser Actor improves average task length from 3.27 to 3.51 (+7.3%). Real-robot experiments further confirm that geometric consistency translates to reliable physical deployment. Cross-architecture validation on OpenVLA-OFT, shows that SE(3) score matching improves LIBERO Long success rate from 92.20 to 94.13, demonstrating that the benefit originates from the Lie formulation itself. Furthermore, our ablations demonstrate that intrinsic geometry and architectural modifications contribute independently to these gains. The primary contributions of this work are summarized as follows:

- We identify and formalize the Euclidean Fallacy in diffusion-based VLA policies, revealing systematic manifold drift and broken equivariance under coordinate transformations.
- We propose Lie Diffuser Actor with three proven guarantees: manifold drift elimination through group closure, left-invariant equivariance for coordinate-frame robustness, and geodesic trajectory generation yielding kinematically optimal screw motions.
- We demonstrate consistent improvements on CALVIN, OpenVLA-OFT, and real-robot experiments, with zero-shot generalization to unseen workspaces confirming the practical value of intrinsic geometric consistency.

## 2. Preliminaries

In this section, we provide the fundamental background knowledge for this literature. Section 2.1 reviews the geometry of SE(3) and introduces the Lie algebraic structure.

Section 2.2 formalizes the policy learning problem and contrasts the Euclidean and the intrinsic perturbation models.

### 2.1. Geometry of SE(3)

**The Special Euclidean Group.** A rigid-body configuration in three-dimensional space is represented by an element of the Special Euclidean group, which is defined as follows:

$$SE(3) = \left\{ \begin{pmatrix} R & \mathbf{t} \\ \mathbf{0}^\top & 1 \end{pmatrix} : R \in SO(3), \mathbf{t} \in \mathbb{R}^3 \right\}, \quad (1)$$

where $SO(3) = \{R \in \mathbb{R}^{3\times3} : R^\top R = I, \det(R) = 1\}$ is the Special Orthogonal group representing all valid 3D rotations. Each element $g \in SE(3)$ is a $4 \times 4$ homogeneous transformation matrix encoding both a rotation $R$ and a translation vector $\mathbf{t}$. The group operation corresponds to sequential rigid transformations: applying transformation $(R_1, \mathbf{t}_1)$ followed by $(R_2, \mathbf{t}_2)$ yields the composition $(R_1, \mathbf{t}_1) \cdot (R_2, \mathbf{t}_2) = (R_1 R_2, R_1 \mathbf{t}_2 + \mathbf{t}_1)$. Geometrically, $SE(3)$ forms a six-dimensional Lie group and a curved Riemannian manifold, meaning it locally resembles Euclidean space but has global curvature. Standard Euclidean operations such as vector addition are geometrically undefined on this manifold. Adding two rotation matrices does not generally produce a valid rotation, and averaging poses in the ambient space yields configurations that violate the orthogonality constraint $\mathbf{R}^\top \mathbf{R} = \mathbf{I}$. This non-linear geometric structure imposes fundamental constraints on probabilistic modeling and requires manifold-aware diffusion techniques.

**Lie Algebra and Exponential Map.** The Lie algebra $\mathfrak{se}(3)$ is the tangent space to the manifold $SE(3)$ at the identity element, providing a linearized representation of the group structure. Elements of $\mathfrak{se}(3)$ are called *twists* and represent infinitesimal rigid motions. A twist $\boldsymbol{\xi} = (\boldsymbol{\omega}, \mathbf{v}) \in \mathbb{R}^6$ consists of an angular velocity component $\boldsymbol{\omega} \in \mathbb{R}^3$ describing rotation and a linear velocity component $\mathbf{v} \in \mathbb{R}^3$ describing instantaneous translation. Unlike curved manifold $SE(3)$, the Lie algebra $\mathfrak{se}(3)$ is a flat vector space where standard linear operations are well-defined, making it the natural domain for adding Gaussian noise in diffusion models.

The exponential map $\exp : \mathfrak{se}(3) \to SE(3)$ provides crucial bridge between the flat tangent space and the curved manifold, transforming twists into finite rigid transformations:

$$\exp(\boldsymbol{\xi}) = \begin{pmatrix} \exp_{SO(3)}(\boldsymbol{\omega}) & V(\boldsymbol{\omega})\mathbf{v} \\ \mathbf{0}^\top & 1 \end{pmatrix}, \quad (2)$$

where $\exp_{SO(3)}(\boldsymbol{\omega}) = I + \frac{\sin\theta}{\theta}[\boldsymbol{\omega}]_\times + \frac{1-\cos\theta}{\theta^2}[\boldsymbol{\omega}]_\times^2$ is the Rodrigues formula with $\theta = \|\boldsymbol{\omega}\|$ representing the rotation angle, $[\cdot]_\times$ denotes the skew-symmetric matrix operator (i.e., $[\boldsymbol{\omega}]_\times \mathbf{x} = \boldsymbol{\omega} \times \mathbf{x}$ for any vector $\mathbf{x}$), and $V(\boldsymbol{\omega}) = I + \frac{1-\cos\theta}{\theta^2}[\boldsymbol{\omega}]_\times + \frac{\theta-\sin\theta}{\theta^3}[\boldsymbol{\omega}]_\times^2$ is left Jacobian of $SO(3)$ that accounts for the coupling between rotation and

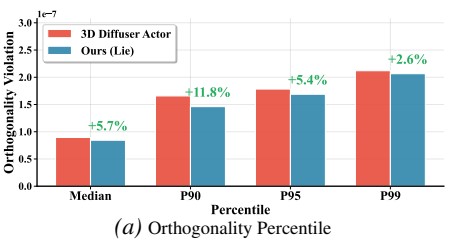

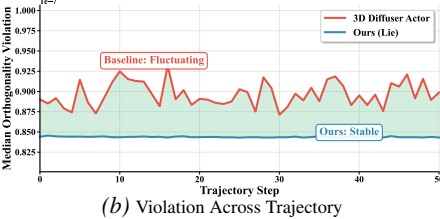

*Figure 2.* SO(3) constraint violation analysis comparing 3D Diffuser Actor and our Lie group representation. (a) Orthogonality violation across percentiles. (b) Median violation across trajectory steps. Our method (blue) maintains stable, low violations.

*(a)* Orthogonality Percentile  *(b)* Violation Across Trajectory

translation. Geometrically, $\exp(\boldsymbol{\xi})$ yields the finite transformation obtained by integrating a constant-velocity motion with twist for unit time, producing a screw motion that combines rotation on an axis with translation along that axis.

A fundamental property exploited throughout this work is that the exponential map is *surjective*: for any twist $\boldsymbol{\xi} \in \mathfrak{se}(3)$, the result $\exp(\boldsymbol{\xi})$ is guaranteed to be a valid element of SE(3) satisfying all manifold constraint.

### 2.2. Problem Formulation

The robot receives visual observations $\mathcal{V} = \{I_1, \ldots, I_K\}$ from $K$ cameras and a language instruction $\mathcal{L}$. The policy generates an end-effector trajectory $\mathbf{g} = (g^1, \ldots, g^H) \in \mathrm{SE}(3)^H$ over horizon $H$. We formulate this as learning a conditional generative model $p_\theta(\mathbf{g} \mid \mathcal{V}, \mathcal{L})$ through a denoising diffusion process.

**Euclidean Formulation.** Standard diffusion policies parameterize poses as vectors $\mathbf{x} \in \mathbb{R}^{12}$ (flattened rotation matrix plus translation) and define the forward process as follows:

$$\mathbf{x}_t = \mathbf{x}_0 + \sigma_t \boldsymbol{\epsilon}, \quad \boldsymbol{\epsilon} \sim \mathcal{N}(\mathbf{0}, \mathbf{I}), \tag{3}$$

where $\sigma_t$ is a noise schedule. This formulation treats SE(3) as a vector space, inducing two problems: (1) intermediate states $\mathbf{x}_t$ violate SO(3) constraints since adding Gaussian noise to a rotation matrix yields a non-orthogonal matrix with probability one, and (2) the additive noise distribution does not transform covariantly under coordinate changes.

**Intrinsic Formulation.** We define diffusion directly on the manifold via left-invariant SDEs. For $g_t \in \mathrm{SE}(3)$:

$$g_t = g_0 \cdot \exp(\sigma_t \boldsymbol{\xi}), \quad \boldsymbol{\xi} \sim \mathcal{N}(\mathbf{0}, \mathbf{I}_6). \tag{4}$$

This multiplicative perturbation model ensures for all $t$ by construction (Proposition 4.1), as the exponential map guarantees valid outputs and SE(3) is closed under multiplication. The left-invariant structure further induces coordinate-frame equivariance (Theorem 4.2): the score function transforms covariantly under rigid actions on the workspace, ensuring that the learned policy is a property of the task rather than of its coordinate representation.

## 3. Geometric Cost of Euclidean Embeddings

The dominant paradigm in diffusion-based manipulation policies treats SE(3) poses as vectors in $\mathbb{R}^n$ and applies additive Gaussian noise during the forward process. This

section examines whether such approximation incurs measurable costs through motivational experiments that reveal geometric errors with consequences for manipulation.

We begin by investigating rotation constraint violation. For any valid rotation matrix $R \in \mathrm{SO}(3)$, the orthogonality condition $R^\top R = I$ must hold exactly. We quantify deviations through the orthogonality error $\epsilon_{\mathrm{orth}} = \|R^\top R - I\|_F$ and compare 3D Diffuser Actor (Ke et al., 2024), which predicts 9D rotation matrices with post-hoc SVD orthogonalization, against our method, which predicts in $\mathfrak{so}(3)$ and maps to SO(3) via the exponential map. Fig. 2 (a) presents a percentile-wise comparison of orthogonality violations across 148K and 123K pose predictions for the baseline and our method, respectively. While both methods operate near float32 numerical precision, our approach achieves consistently lower violations: 5.7% reduction at median, 11.8% at P90, 5.4% at P95, and 2.6% at P99. The temporal structure shown in Fig. 2 (b) reveals further distinctions. The baseline exhibits substantial variance across trajectory steps throughout the 50-step trajectory. This instability arises from SVD projection artifacts that vary with the input matrix condition number. In contrast, our method maintains near-constant violations with minimal variance, as the exponential map guarantees that any $\boldsymbol{\omega} \in \mathfrak{so}(3)$ maps to a valid rotation by construction. These results motivate the core design choice of operating directly on the Lie algebra: predictions in $\mathfrak{so}(3)$ achieve both consistently lower constraint violations and stable behavior, properties essential for generating smooth and physically realizable robot manipulation trajectories. The full framework extending this insight to trajectory-level diffusion on SE(3) is presented in the following section.

## 4. Methodology

This section presents Lie Diffuser Actor, a framework that reformulates diffusion to operate natively on SE(3). We first develop the theoretical foundation in Section 4.1, then describe the neural architecture in Section 4.2.

### 4.1. Intrinsic Diffusion on the SE(3) Manifold

#### 4.1.1. DIFFUSION VIA LEFT-INVARIANT SDEs

Standard diffusion defines forward noising as $\mathbf{x}_t = \mathbf{x}_0 + \sigma_t \boldsymbol{\epsilon}$ with $\boldsymbol{\epsilon} \sim \mathcal{N}(\mathbf{0}, \mathbf{I})$. This formulation is invalid for SE(3) as vector addition is undefined on manifolds. We instead define the forward process as a stochastic differential equation on

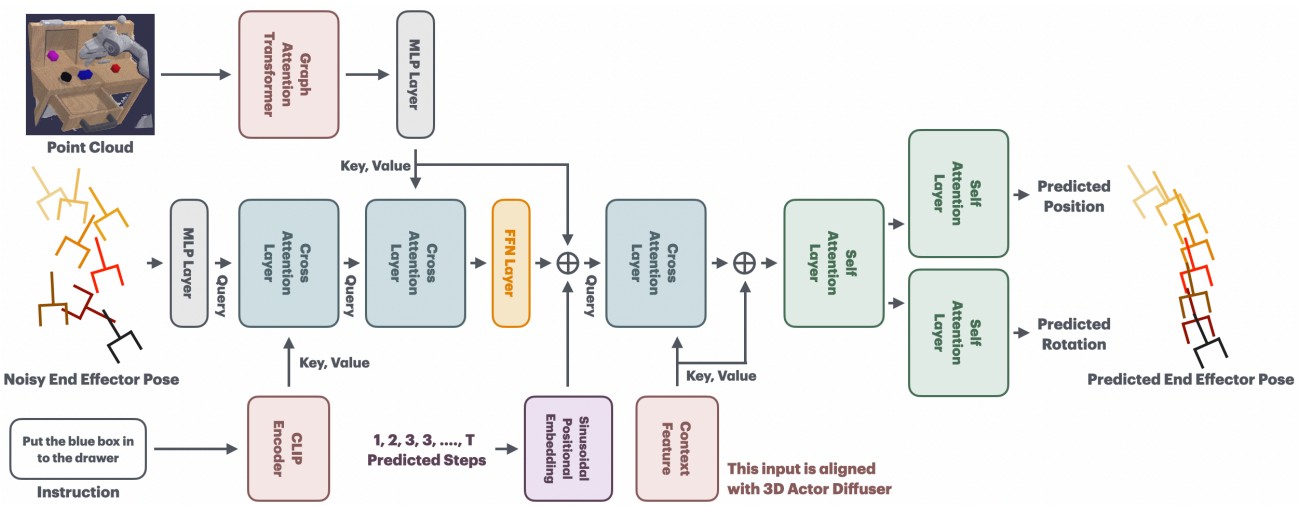

*Figure 3.* The overall architectural details for the proposed Lie Diffuser Actor.

the Lie group. For pose $g_t \in \mathrm{SE}(3)$ evolving over $t \in [0, T]$, the forward diffusion takes the Stratonovich form as follows:

$$\mathrm{d}g_t = g_t \cdot \left( \sigma_t \sum_{i=1}^{6} E_i \circ \mathrm{d}W_t^i \right), \qquad (5)$$

where $\{E_i\}_{i=1}^{6}$ form an orthonormal basis for $\mathfrak{se}(3)$, $\{W_t^i\}$ are independent Wiener processes, and $\sigma_t$ is a time-dependent noise scale. In discrete form, this reduces to $g_t = g_0 \cdot \exp(\sigma_t \boldsymbol{\xi})$ with $\boldsymbol{\xi} \sim \mathcal{N}(\mathbf{0}, \mathbf{I}_6)$, where $\mathfrak{se}(3) \to \mathrm{SE}(3)$ denotes the Lie group exponential map (Appendix A.2.1).

This construction addresses a fundamental issue with Euclidean formulations: adding Gaussian noise to a rotation matrix yields a non-orthogonal matrix with probability one. Such manifold drift forces networks to learn correction functions via SVD or quaternion normalization. The left-invariant formulation eliminates this problem entirely.

**Proposition 4.1** (Elimination of Manifold Drift). *The left-invariant SDE in Eq. (5) ensures $g_t \in \mathrm{SE}(3)$ for all $t \in [0, T]$ almost surely.*

The proof, based on Lie group closure under multiplication, is provided in Appendix A.2.1. This property allows the network to focus on learning manipulation semantics rather than geometric corrections.

### 4.1.2. REVERSE PROCESS AND EQUIVARIANCE

For trajectory generation, we require the reverse-time dynamics. Following time-reversal theory on Lie groups (Hsu, 2002), the reverse process satisfies the following equation:

$$\mathrm{d}g_t = g_t \cdot \left( \sigma_t^2 s_\theta(g_t, t) \, \mathrm{d}t + \sigma_t \, \mathrm{d}\bar{\mathbf{B}}_t \right), \qquad (6)$$

where $s_\theta : \mathrm{SE}(3) \times [0, T] \to \mathfrak{se}(3)$ is the learned score function approximating $\nabla_{\mathfrak{se}(3)} \log p_t$, and $\bar{\mathbf{B}}_t$ is reverse-time Brownian motion. For practical implementation, we

discretize via the exponential map retraction: $g_{t-\Delta t} = g_t \cdot \exp(\sigma_t^2 s_\theta(g_t, t) \Delta t + \sigma_t \sqrt{\Delta t} \, \boldsymbol{\zeta})$ with $\boldsymbol{\zeta} \sim \mathcal{N}(\mathbf{0}, \mathbf{I}_6)$.

Beyond preserving geometric validity, the left-invariant structure induces equivariance under coordinate transformations. This property is essential for robotic manipulation, where the choice of reference frame is often arbitrary and may vary across deployments.

**Theorem 4.2** (Left-Invariant Equivariance). *For the optimal score function of Eq. (5) and any rigid transformation $h \in \mathrm{SE}(3)$,*

$$s_\theta(h \cdot g, t) = \mathrm{Ad}_h(s_\theta(g, t)), \qquad (7)$$

*where the adjoint representation acts as $\mathrm{Ad}_{(R, \mathbf{p})}(\boldsymbol{\omega}, \mathbf{v}) = (R\boldsymbol{\omega}, R\mathbf{v} + [\mathbf{p}]_\times R\boldsymbol{\omega})$.*

Theorem 4.2 (proved in Appendix A.3.1) formalizes this result: the score function operates in a body-fixed reference frame, and the learned policy therefore depends on the task geometry rather than on its coordinate representation. In Section 5, the consequences of this equivariance emerge in the SO(3) constraint violations during denoising (Section 5.4), the coordinate-frame-stable refinement of target poses (Section 5.5), and the zero-shot generalization across environment configurations in the CALVIN ABC $\to$ D protocol (Section 5.2).

### 4.1.3. KINEMATIC OPTIMALITY

The final theoretical contribution concerns trajectory quality. The deterministic probability flow ODE corresponding to Eq. (6) is $\frac{\mathrm{d}g_t}{\mathrm{d}t} = g_t \cdot \sigma_t^2 s_\theta(g_t, t)$, which connects our formulation to Riemannian geometry.

**Proposition 4.3** (Geodesic Trajectories). *When the score function is constant along the trajectory, i.e., $s_\theta(g_t, t) = \boldsymbol{\xi}^*$ for fixed $\boldsymbol{\xi}^* \in \mathfrak{se}(3)$, the probability flow generates geodesics on $\mathrm{SE}(3)$ under the bi-invariant metric. These geodesics correspond to screw motions with constant angular and linear velocities.*

The proof is provided in Appendix A.4.1. In practice, the learned score varies along trajectories, but the intrinsic formulation biases toward geodesic-like behavior, resulting in smoother motion with reduced angular jerk compared to Euclidean baselines.

## 4.2. Neural Architecture

Building on the theoretical framework established in Section 4.1, we now describe the neural architecture that instantiates intrinsic SE(3) diffusion for robot manipulation. The design extends the 3D Diffuser Actor (Ke et al., 2024) with principled modifications to the prediction head and training objective that respect the manifold structure of rigid body transformations. Figure 3 illustrates the overall architecture, highlighting the flow from multimodal observations through geometric encoding, iterative denoising, and manifold-aware prediction.

### 4.2.1. GEOMETRIC CONTEXT ENCODING

The encoder extracts task-relevant features from multimodal observations while preserving the inherent 3D geometric structure of the scene. RGB-D observations from $K$ cameras positioned around the workspace are back-projected into a unified point cloud representation using camera intrinsics and extrinsics, yielding a dense 3D scene reconstruction. This point cloud is then processed by a graph attention transformer (Veličković et al., 2018) to obtain spatially-grounded geometric features $\mathbf{F}_{\text{geo}} \in \mathbb{R}^{N \times d}$, where $N$ represents the number of non-empty voxels and $d$ is the feature dimension. The sparse convolution architecture is particularly well-suited for processing point clouds as it operates efficiently on the spatially sparse data while preserving the metric relationships between the 3D locations and capturing both local geometric details and global scene structure.

In parallel, natural language instructions describing the manipulation task are encoded via a pretrained CLIP (Radford et al., 2021) text encoder, yielding language features $\mathbf{F}_{\text{lang}} \in \mathbb{R}^{L \times d}$ where $L$ is the number of text tokens. The use of a frozen pretrained language model leverages large-scale vision-language pretraining to provide robust semantic understanding without requiring task-specific language supervision. Cross-attention layers then fuse these complementary modalities, allowing the model to ground linguistic concepts in specific spatial locations within the geometric representation. This produces a unified context representation $\mathcal{C}$ that conditions all subsequent denoising steps, ensuring that predicted trajectories are both geometrically feasible and semantically aligned with the instruction.

[1]Retrained at 600K iterations post-submission to match the baseline training budget; the originally reported 400K result was 3.254.

### 4.2.2. ITERATIVE DENOISING TRANSFORMER

The denoising backbone is a Transformer (Vaswani et al., 2017) that refines noisy pose trajectories through repeated application of the learned score function. At each denoising step indexed by diffusion time $t \in \{T, T-1, \ldots, 1\}$, the Transformer receives the current trajectory estimate $\mathbf{g}_t = (g_t^1, \ldots, g_t^H) \in \text{SE}(3)^H$ consisting of $H$ waypoints, along with a sinusoidal time embedding $\tau(t) \in \mathbb{R}^{d_t}$ that encodes the current noise level. Each pose $g_t^h = (R_t^h, \mathbf{t}_t^h)$ is tokenized into a feature vector by concatenating learned positional embeddings for the translation $\mathbf{t}_t^h$, axis-angle representations of the rotation $R_t^h$ processed through dedicated MLPs, and binary embeddings indicating gripper state (open/closed). This tokenization scheme preserves the SE(3) structure while enabling efficient parallel processing.

Self-attention layers within the Transformer model temporal dependencies across the action horizon $H$, allowing the network to capture multi-step coordination and sequential constraints (e.g., the gripper must open before grasping, and the object must be grasped before placement). Cross-attention layers attend to the geometric-linguistic context $\mathcal{C}$ at each denoising step, dynamically injecting task-relevant spatial and semantic information that guides the trajectory refinement. This architecture mirrors the iterative denoising process of diffusion models while respecting the temporal structure of robot trajectories, enabling the model to progressively refine coarse motion plans into precise actions.

### 4.2.3. TANGENT SPACE PREDICTION HEAD

The critical architectural modification distinguishing our approach from standard Euclidean diffusion models is the reformulation of the output head to predict score vectors in the Lie algebra $\mathfrak{se}(3)$ rather than ambient-space noise. For each trajectory waypoint $h \in \{1, \ldots, H\}$, the prediction head outputs a 6-dimensional twist $\boldsymbol{\xi}^h = (\boldsymbol{\omega}^h, \mathbf{v}^h) \in \mathbb{R}^6$ via separate multi-layer perceptrons for the angular velocity component $\boldsymbol{\omega}^h \in \mathbb{R}^3$ and linear velocity component $\mathbf{v}^h \in \mathbb{R}^3$. This factorization reflects the semi-direct product structure of SE(3) and allows the network to learn rotation and translation dynamics with appropriate inductive biases.

Critically, the predicted twist $\boldsymbol{\xi}^h$ lives in the flat tangent space $\mathfrak{se}(3)$ where Gaussian noise is well-defined, but the denoising update is performed on the manifold via the exponential map: $g_{t-1}^h = g_t^h \cdot \exp(-\beta_t \boldsymbol{\xi}^h)$, where $\beta_t$ is the noise schedule. The exponential map $\exp : \mathfrak{se}(3) \to \text{SE}(3)$ guarantees by construction that each update yields a valid rigid body transformation satisfying all manifold constraints $(R^T R = I, \det(R) = 1)$, eliminating the need for post-hoc projection or renormalization. This design choice directly implements the manifold-preserving property established in Proposition 4.1, ensuring geometric validity throughout

*Table 1.* Comparison for 3D Diffuser Actor and our method with various settings on CALVIN benchmark.

| Method | SR1 | SR2 | SR3 | SR4 | SR5 | Average Length |
|---|---|---|---|---|---|---|
| **ABC → D** | | | | | | |
| 3D Diffuser Actor (600K) | 92.2 | 78.7 | 63.9 | 51.2 | 41.2 | 3.27 |
| **Lie Diffuser Actor (600K) w/o** GAT Encoder[1] | 89.6 | 78 | 66.6 | 55.7 | 46.9 | 3.368 |
| **Lie Diffuser Actor (300K) w/o** Lie Space Diffusion | 90.2 | 80.3 | 69.6 | 58.5 | 48.8 | 3.474 |
| **Lie Diffuser Actor (300K)** | **93.7** | **83.4** | **70.3** | **57.6** | **46.2** | **3.512** |
| **ABCD → D** | | | | | | |
| 3D Diffuser Actor (800K) | 90.3 | 77.3 | 65.8 | 53.8 | 41.6 | 3.288 |
| **Lie Diffuser Actor (300K) w/o** GAT Encoder | 90.8 | 77.3 | 66.4 | 57.6 | 48.3 | 3.404 |
| **Lie Diffuser Actor (400K) w/o** Lie Space Diffusion | 91.0 | 76.1 | 63.4 | 51.6 | 41.8 | 3.239 |
| **Lie Diffuser Actor (300K)** | **90.6** | **80.4** | **71.1** | **62.6** | **53.7** | **3.584** |

the entire sampling process. Gripper actions are predicted via a separate sigmoid classifier head that outputs binary open/close commands, which are synchronized with the corresponding pose waypoints during trajectory execution.

### 4.2.4. TRAINING OBJECTIVE

The score function is trained via denoising score matching in the Lie algebra $\mathfrak{se}(3)$, which provides a principled framework for learning the reverse diffusion process while respecting the manifold structure. Given a ground-truth demonstration trajectory $\mathbf{g}_0 = (g_0^1, \ldots, g_0^H)$ consisting of $H$ waypoints, we construct training samples by first uniformly sampling a diffusion timestep $t \sim \mathcal{U}(0, T)$ and independent Gaussian noise twists $\boldsymbol{\xi}^h \sim \mathcal{N}(\mathbf{0}, \mathbf{I}_6)$ for each waypoint $h$. The noisy poses are then generated via the forward diffusion process using the exponential map: $g_t^h = g_0^h \cdot \exp(\sigma_t \boldsymbol{\xi}^h)$, where $\sigma_t$ is the noise schedule that controls the magnitude of perturbation at timestep $t$. This construction ensures that noise is added intrinsically through the manifold's exponential map rather than through ambient space addition, preserving the SE(3) structure through the forward process.

The objective combines three loss terms that jointly supervise pose prediction, translation accuracy, and gripper state:

$$\mathcal{L} = \lambda_s \mathbb{E}_{t,\boldsymbol{\xi}} \left[ \sum_{h=1}^{H} \| s_\theta(g_t^h, t) - \boldsymbol{\xi}^h \|^2 \right] + \lambda_p \mathcal{L}_{\text{pos}} + \lambda_g \mathcal{L}_{\text{grip}}, \tag{8}$$

where $s_\theta(g_t^h, t)$ denotes the network's predicted twist for waypoint $h$ at diffusion time $t$. The primary score matching term (weighted by $\lambda_s$) trains the network to predict the noise twist $\boldsymbol{\xi}^h$ added during the forward process, directly implementing denoising score matching adapted to the manifold setting. The auxiliary position loss $\mathcal{L}_{\text{pos}}$ applies standard mean squared error on translation components to provide additional supervision for spatial accuracy, while the gripper

loss $\mathcal{L}_{\text{grip}}$ utilizes binary cross-entropy to classify gripper states (i.e., open/closed) at each waypoint. The loss weights $\lambda_s$, $\lambda_p$, and $\lambda_g$ balance these objectives during training.

## 5. Experimental Results

### 5.1. Experimental Setups

**Benchmarks.** We evaluate the proposed Lie Diffuser Actor on CALVIN (Mees et al., 2022), a long-horizon language-conditioned benchmark requiring policies to execute up to five consecutive tasks from natural language instructions. The benchmark defines 34 manipulation skills across four environment variations (A, B, C, D) with distinct object layouts and appearances. Following the standard ABC→D protocol, we train on environments A, B, C and evaluate zero-shot transfer to the held-out environment D. We report the success rates SR1–SR5 and average task chain length.

**Baselines.** This work investigates geometric representations for diffusion-based action generation, orthogonal to advances in vision-language foundation models. We therefore compare against methods within the same architectural class rather than large-scale VLA models trained on orders-of-magnitude more data. Our primary baseline is 3D Diffuser Actor (Ke et al., 2024), the state-of-the-art trajectory-level diffusion policy that shares our 3D point cloud encoder paradigm but employs Euclidean SE(3) parameterization. To disentangle contributions from intrinsic geometry versus architectural modifications, we conduct four controlled ablations: (i) the original 3D Diffuser Actor with Euclidean diffusion as the baseline, (ii) 3D Diffuser Actor with our Lie group diffusion to isolate the effect of intrinsic geometry, (iii) 3D Diffuser Actor with a Graph Attention Network (GAT) encoder to isolate the effect of architectural improvements, and (iv) Lie Diffuser Actor combining both modifications. This factorial design enables precise attribution of performance gains to each component.

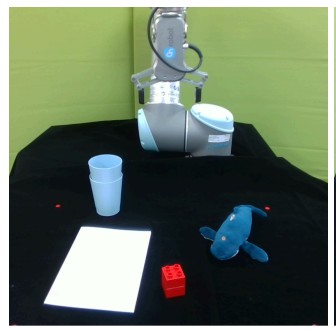
*(a)* Move Doll Platform

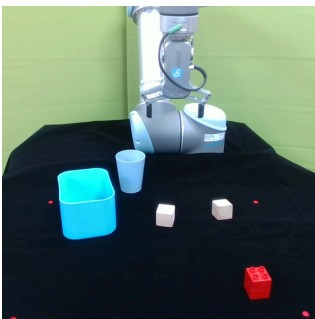
*(b)* Put Block in Box

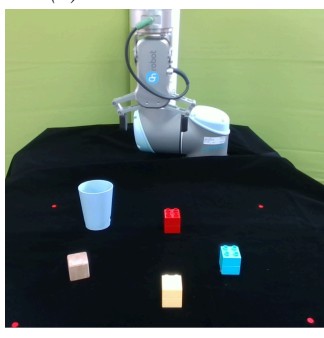
*(c)* Sort Blocks

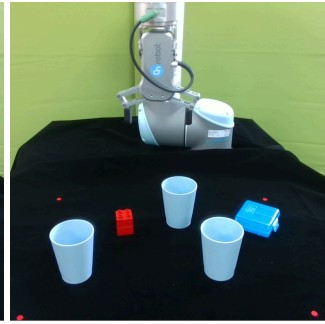
*(d)* Stack Cups

*Figure 4.* Real robot evaluation environments.
*Table 2.* Real robot success rates (%) over 20 trials per task.

| Method | Move Doll | Block in Box | Sort Blocks | Stack Cups |
|---|---|---|---|---|
| Baseline | 90 | 80 | 55 | 55 |
| Lie Diffuser | **100** | 75 | **75** | **60** |

## 5.2. Long-Horizon Manipulation Performance

Table 1 presents results on the zero-shot ABC→D transfer task, which assesses the impact of geometric consistency on long-horizon planning. The proposed contributions, intrinsic Lie group diffusion and the geometry-aware GAT encoder, prove individually effective rather than merely complementary. The ablation results demonstrate that augmenting the baseline's 3D relative attention with our GAT encoder improves performance by better capturing local geometric structure in point clouds. Similarly, replacing only the Euclidean diffusion with Lie group diffusion improves performance by eliminating manifold drift and ensuring coordinate-frame equivariance. The full Lie Diffusor Actor methodology achieves the highest success rates and chain lengths, confirming that combining both components is essential for trajectory consistency in long-horizon tasks.

The lower section of Table 1 reports the ABCD→D setting, where models train on all four environments to test robustness under increased environment diversity. The Euclidean baseline exhibits training instability and limited generalization on this diverse dataset, whereas our approach

demonstrates superior scalability. We attribute this to the baseline's sensitivity to coordinate-frame variations: diverse environments introduce greater variation in object configurations and workspace layouts, which exacerbates the coordinate-frame dependence of Euclidean parameterization. The ablated variants of our methodology still surpass baseline performance even under same training budget despite incorporating only one of the two proposed components, which are consistent with the zero-shot findings. The full LDA methodology yields the best performance, as geometric consistency prevents compounding errors in complex, multi-stage manipulation.

## 5.3. Real Robot Validation

We deployed both the baseline and our method on a real robot arm across four manipulation tasks that span coarse transport to fine-grained assembly, as shown in Fig. 4: (i) **Move Doll Platform** requires stable 6-DOF transport to prevent tipping, (ii) **Put Block in Box** tests tight-tolerance insertion under noisy depth sensing, (iii) **Sort Blocks** demands accurate spatial discrimination and precise translational placement, and (iv) **Stack Cups** requires sub-centimeter alignment for stability. Detailed experiment settings are provided in Appendix D. We collected 50 demonstrations per task with a two-stage protocol. The first stage uses human-guided kinesthetic teaching to capture natural 6-DOF trajectories. The second stage performs autonomous replay to record clean visual observations without human occlusions. This protocol ensures high-quality training data free from spurious visual correlations.

The four tasks engage the SE(3) structure of the action space along a spectrum from rotation-dominated to translation-dominated coupling. Move Doll Platform and Put Block in Box require orientation invariance throughout translation, since the end-effector must hold a consistent forward-facing pose during transport that keeps the carried object from drifting away from its target configuration. Stack Cups concentrates the demand at the insertion stage, where a precise tilted approach posture provides the clearance needed for the upper cup to align with the rim of the lower one. Sort Blocks operates under a single target orientation and therefore shifts the burden from rotational range to geometric scene perception, which calls for accurate understanding of the relative configuration of foreground and background blocks. Across this spectrum, the first three tasks couple orientation and translation to varying degrees, and it is precisely in this coupled regime that Euclidean interpolation departs most severely from valid SE(3) trajectories (Proposition 4.3).

As shown in Table 2, Lie Diffuser Actor achieves superior performance on tasks that demand high geometric fidelity. The method attains perfect success on **Move Doll Platform** and substantial improvements on **Sort Blocks** and **Stack**

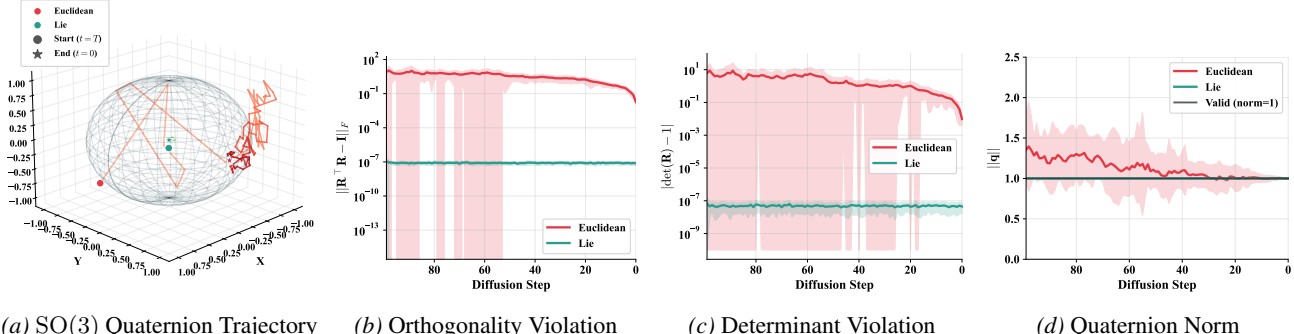

*(a)* SO(3) Quaternion Trajectory     *(b)* Orthogonality Violation     *(c)* Determinant Violation     *(d)* Quaternion Norm

*Figure 5.* **Manifold constraint analysis during reverse diffusion.** (a) Quaternion trajectories projected onto $\mathbb{S}^3$ for Euclidean (red) and Lie (teal) diffusion. Circle markers denote the starting point ($t = T$, pure noise), and star markers denote the final denoised output ($t = 0$). The Euclidean trajectory departs from the unit sphere, while the Lie trajectory remains on the manifold. (b) Orthogonality violation $\|\mathbf{R}^\top \mathbf{R} - \mathbf{I}\|_F$ over diffusion steps. (c) Determinant violation $|\det(\mathbf{R}) - 1|$ over diffusion steps. (d) Quaternion norm $\|\mathbf{q}\|$ over diffusion steps, where valid rotations require $\|\mathbf{q}\| = 1$. Shaded regions indicate $\pm 1$ standard deviation across multiple samples. The proposed Lie Diffuser Actor maintains geometric validity throughout, while Euclidean diffusion violates SO(3) constraints by 7+ orders of magnitude.

**Cups**, where precise orientation control is critical. These gains confirm that Lie group diffusion effectively mitigates geometric drift in real-world settings, where sensor noise and actuation errors compound over trajectory execution. For simpler insertion tasks such as **Put Block in Box**, LDA remains comparable to the baseline, where unconstrained Euclidean exploration may offer a slight advantage.

### 5.4. Diffusion Manifold Comparison

Euclidean diffusion models fail to respect the geometric structure of SO(3) when generating rotation actions. We empirically demonstrate this limitation by visualizing denoising trajectories and quantifying manifold constraint violations at each timestep. For both the Euclidean baseline and our Lie diffuser, we record intermediate rotation predictions during the reverse diffusion process (from $t = T$ to $t = 0$). The Euclidean model outputs are saved *before* any post-hoc Gram-Schmidt orthogonalization, while the Lie model outputs are obtained via the exponential map.

**Qualitative Analysis.** For visualization, we convert rotation matrices to unit quaternions and project them onto the unit sphere $\mathbb{S}^3$. As shown in Fig. 5 (a), the Euclidean diffusion trajectory (red) exhibits erratic behavior and frequently departs from the unit sphere surface, particularly during the early noisy stages ($t \approx T$). This manifests as the trajectory cutting *through* the interior of the sphere rather than traversing along its surface. In contrast, the Lie Diffuser Actor's trajectory (teal) remains precisely on the manifold throughout the entire denoising process and follows geodesic paths that faithfully respect the curved geometry of rotation space.

**Quantitative Analysis.** We measure two key SO(3) constraint violations: (i) *Orthogonality Error* $\|\mathbf{R}^\top \mathbf{R} - \mathbf{I}\|_F$, and (ii) *Determinant Error* $|\det(\mathbf{R}) - 1|$. As shown in Figs. 5 (b-c), the Euclidean baseline exhibits substantial

violations throughout the diffusion process, with errors reaching $\mathcal{O}(10^0)$, while the Lie diffuser maintains errors at floating-point precision ($\sim 10^{-7}$). We also track the quaternion norm $\|\mathbf{q}\|$ during denoising in Fig. 5(d). Valid unit quaternions require $\|\mathbf{q}\| = 1$, but Euclidean outputs deviate significantly with values ranging from $0.5$ to $2.0$. The proposed Lie Diffuser Actor maintains unit norm throughout and eliminates the requirement for post-hoc normalization.

These results show that the geometric constraints are not only a theoretical concern. They manifest as measurable violations during inference. Post-hoc projection, the standard remedy, introduces three compounding drawbacks, all rooted in its application at inference time alone. First, it creates a training-inference mismatch, in which the network trains on unnormalized intermediate states and faces evaluation only after projection. Second, it amplifies error through the sensitivity of SVD to near-degenerate matrices, where small prediction errors translate into disproportionately large changes in the projected rotation. Third, it breaks generative consistency through a non-differentiable inference-time projection that alters the reverse-time SDE in a manner the training objective never accounts for. Our Lie formulation avoids all three at once by operating natively on the tangent space, where every step is geometrically valid by construction and no projection is ever required.

### 5.5. Denoising Stability and Look-ahead Consistency

To investigate the internal consistency of the denoising process, we analyze the stability of the look-ahead prediction $\hat{x}_0^{(t)}$, which represents the model's estimate of the final clean action at each timestep $t$. We define Geodesic Jitter as the angular distance (in degrees) between consecutive look-ahead predictions: $d_{\text{geo}}(\hat{x}_0^{(t)}, \hat{x}_0^{(t-1)})$. As illustrated in Fig. 6, the Euclidean baseline exhibits nearly an order of magnitude higher jitter than LDA throughout the reverse diffusion pro-

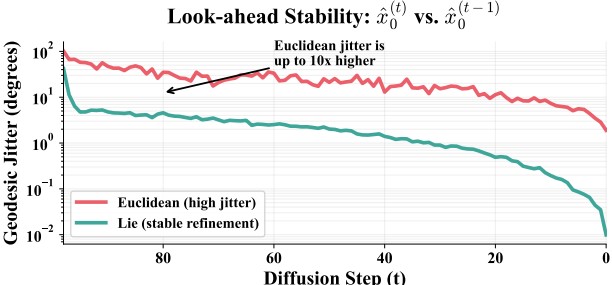

*Figure 6.* **Look-ahead Stability and Geodesic Jitter.** Geodesic jitter (angular distance in degrees) between consecutive look-ahead predictions $\hat{x}_0^{(t)}$ and $\hat{x}_0^{(t-1)}$ during reverse diffusion. The Euclidean baseline (red) exhibits up to two orders of magnitude higher jitter than LDA (teal), particularly in early diffusion steps.

cess. This instability is a direct consequence of manifold drift inherent to Euclidean methods. Unconstrained updates in $\mathbb{R}^{12}$ do not respect $SO(3)$ curvature, and the subsequent orthogonalization required to restore validity amplifies small network errors into large, erratic jumps in rotation space. In contrast, Lie Diffuser Actor diffuses natively on the manifold and achieves smooth, monotonic refinement of the target action. This stability is critical for robotics applications: the model's estimate of the final goal evolves predictably, which produces consistent control signals and prevents jerky behavior associated with unconstrained rotation diffusion.

## 6. Related Works

Diffusion-based VLA policies (Chi et al., 2023; Urain et al., 2023; Ke et al., 2024; Ze et al., 2024; Liu et al., 2024; Cheng et al., 2024) have established trajectory-level denoising as the dominant paradigm for robotic manipulation, yet uniformly parameterize SE(3) poses as flat Euclidean vectors and rely on post-hoc projections to enforce geometric validity. Equivariant policy learning (Yang et al., 2024; Ryu et al., 2024; Tie et al., 2025; Zhu et al., 2025) encodes geometric structure in neural architectures but restricts equivariance to the encoder while the generative process remains Euclidean, creating a fundamental asymmetry. Riemannian generative models (De Bortoli et al., 2022; Lou et al., 2023; Chen & Lipman, 2024; Braun et al., 2024) provide mathematical foundations for manifold-valued generation, though flow matching constructs deterministic transport maps that may inadequately capture the multimodality inherent in manipulation. Lie Diffuser Actor synthesizes these directions: from diffusion policies, we inherit trajectory-level generation; from equivariant learning, explicit geometric encoding; from Riemannian methods, intrinsic manifold operations. Our left-invariant SDE provides stronger equivariance guarantees than general Riemannian approaches while maintaining stochastic dynamics essential for multimodal action distribution. Extended discussion appears in Appendix B.

## 7. Conclusions

In this paper, we introduced Lie Diffuser Actor, which addresses geometric inconsistencies in manipulation policies by operating intrinsically on the SE(3) manifold. By formulating diffusion through left-invariant SDEs, our method eliminates manifold drift, ensures coordinate-frame equivariance, and generates kinematically optimal geodesic trajectories. Both simulation and real-robot experiments validate that this intrinsic formulation leads to significant improvements in long-horizon tasks and precise orientation control. These results confirm that respecting the intrinsic structure of SE(3) is essential for robust physical deployment.

## Acknowledgements

The authors gratefully acknowledge the support from the National Science and Technology Council (NSTC) in Taiwan under grant numbers NSTC 114-2221-E-002-069-MY3, NSTC 113-2221-E-002-212-MY3, and NSTC 114-2218-E-A49-026, as well as the support from the Academia Sinica Scholar Award (ASSA) under grant number AS-ASSA-115-02, NTU Artificial Intelligence Center of Research Excellence, and Taiwan Centers of Excellence in Artificial Intelligence. This research was also supported by the NVIDIA Academic Grant Program. The authors would also like to express their appreciation for the hardware grant donation of the GPUs from NVIDIA Corporation and NVIDIA AI Technology Center (NVAITC) used in this work. Furthermore, the authors extend their gratitude to the National Center for High-Performance Computing (NCHC) for providing computational and storage resources. The authors also thank the NVIDIA Taipei-1 supercomputer for providing essential computing resources.

## Impact Statement

This paper presents a geometrically principled approach to robot manipulation policy learning. The primary societal impact lies in improving the reliability and generalization of robotic systems, which may accelerate automation in manufacturing, logistics, and assistive applications. While increased automation raises legitimate concerns about workforce displacement, we believe that safer and more predictable robot behavior, enabled by methods that respect physical constraints by construction, is a prerequisite for responsible deployment. We do not anticipate unique negative consequences beyond those associated with general advances in robot learning.

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

# A. Proof of Theoretical Framework

This section provides detailed mathematical proofs for the three core theoretical results presented in the main paper. We begin with a comprehensive table of notation to facilitate readability, followed by rigorous derivations establishing manifold preservation, equivariance, and kinematic optimality. Each proof is accompanied by detailed insights that connect the mathematical formalism to the geometric and physical intuition underlying our approach.

## A.1. Notation and Definitions

Throughout the proofs, we adopt the following mathematical conventions and notation. Table 3 summarizes the key symbols used in our theoretical analysis.

*Table 3.* Mathematical notation used in theoretical proofs.

| Symbol | Definition |
|---|---|
| $SE(3)$ | Special Euclidean group; manifold of rigid body poses $(R, t)$ |
| $SO(3)$ | Special orthogonal group; manifold of rotation matrices |
| $\mathfrak{se}(3)$ | Lie algebra of $SE(3)$; tangent space at identity |
| $\mathfrak{so}(3)$ | Lie algebra of $SO(3)$; space of skew-symmetric matrices |
| $g_t$ | Pose element at diffusion timestep $t$, where $g_t \in SE(3)$ |
| $R$ | Rotation matrix, $R \in SO(3)$ |
| $t$ | Translation vector, $t \in \mathbb{R}^3$ (also used for time index) |
| $\xi = (\omega, v)$ | Velocity twist in $\mathfrak{se}(3)$ where $\omega, v \in \mathbb{R}^3$ |
| $\omega$ | Angular velocity component of twist |
| $v$ | Linear velocity component of twist |
| $\exp(\cdot)$ | Exponential map $\exp : \mathfrak{se}(3) \to SE(3)$ |
| $\log(\cdot)$ | Logarithm map $\log : SE(3) \to \mathfrak{se}(3)$ |
| $[\omega]_\times$ | Skew-symmetric matrix associated with vector $\omega$ |
| $\hat{\xi}$ | Matrix representation of twist $\xi$ in $\mathfrak{se}(3)$ |
| $\mathrm{Ad}_g$ | Adjoint representation of $g$ on the Lie algebra |
| $s_\theta(g, t)$ | Learned score function predicting twist in $\mathfrak{se}(3)$ |
| $\beta_t$ | Time-dependent noise schedule |
| $dW_t$ | Standard Brownian motion in $\mathbb{R}^6$ |
| $\sigma_t$ | Noise scale at time $t$ in Euclidean formulation |
| $\epsilon$ | Random Gaussian noise vector |
| $p_t(g)$ | Probability density at time $t$ |
| $\nabla_g$ | Riemannian gradient operator on manifold |
| $\| \cdot \|_F$ | Frobenius norm for matrices |
| $\| \cdot \|_{\mathfrak{se}(3)}$ | Bi-invariant norm on Lie algebra |
| $\langle \cdot, \cdot \rangle$ | Inner product on Lie algebra |
| $L_h$ | Left-multiplication by group element $h$ |
| $\mathcal{L}(\theta)$ | Loss functional for score function parameters |
| $V(\omega)$ | Left Jacobian of $SO(3)$ |
| $h$ | Arbitrary group element for transformations |
| $\delta t$ | Infinitesimal time step |

We further establish several conventions used throughout the proofs. Matrix elements are denoted with subscripts, e.g., $\epsilon_{ij}$ for the $(i, j)$-th entry of matrix $\epsilon$. The trace of a matrix $M$ is denoted $\mathrm{tr}(M)$, and the determinant is $\det(M)$. For a vector $\omega = (\omega_1, \omega_2, \omega_3)^T \in \mathbb{R}^3$, the associated skew-symmetric matrix is

$$[\omega]_\times = \begin{bmatrix} 0 & -\omega_3 & \omega_2 \\ \omega_3 & 0 & -\omega_1 \\ -\omega_2 & \omega_1 & 0 \end{bmatrix}. \tag{9}$$

The homogeneous matrix representation of an element $g = (R, t) \in SE(3)$ is

$$g = \begin{bmatrix} R & t \\ 0 & 1 \end{bmatrix} \in \mathbb{R}^{4 \times 4}. \tag{10}$$

Similarly, a twist $\xi = (\omega, v) \in \mathfrak{se}(3)$ has matrix representation

$$\hat{\xi} = \begin{bmatrix} [\omega]_\times & v \\ 0 & 0 \end{bmatrix} \in \mathbb{R}^{4 \times 4}. \tag{11}$$

## A.2. Elimination of Manifold Drift

### A.2.1. PROOF AND DERIVATION

*Proof.* We first establish the manifold violation in the Euclidean case. Write $g = (R, t)$ where $R \in \mathbb{R}^{3 \times 3}$ and $t \in \mathbb{R}^3$. Under the Euclidean update, the rotation component evolves as $R_t = R_0 + \sigma_t \epsilon_R$ where $\epsilon_R \in \mathbb{R}^{3 \times 3}$ has independent Gaussian entries. This formulation treats the rotation matrix as if it were an unconstrained element of the ambient Euclidean space $\mathbb{R}^{3 \times 3}$, completely ignoring the geometric structure that defines valid rotations. For $R_t$ to lie in $SO(3)$, we require $R_t^T R_t = I$ and $\det(R_t) = +1$. These constraints encode the fundamental properties that rotations preserve lengths and orientations. Computing the orthogonality condition yields

$$R_t^T R_t = (R_0 + \sigma_t \epsilon_R)^T (R_0 + \sigma_t \epsilon_R) = I + \sigma_t (R_0^T \epsilon_R + \epsilon_R^T R_0) + \sigma_t^2 \epsilon_R^T \epsilon_R. \tag{12}$$

This expansion reveals the source of the manifold violation. The initial rotation $R_0$ satisfies $R_0^T R_0 = I$ by assumption, but the addition of Gaussian noise $\sigma_t \epsilon_R$ introduces both first-order and second-order deviation terms. The deviation from the identity matrix is $\Delta := R_t^T R_t - I = \sigma_t (R_0^T \epsilon_R + \epsilon_R^T R_0) + \sigma_t^2 \epsilon_R^T \epsilon_R$. The first-order term $\sigma_t (R_0^T \epsilon_R + \epsilon_R^T R_0)$ is symmetric by construction, and represents the linear departure from orthogonality induced by the random perturbation. Since $\epsilon_R$ has independent standard Gaussian entries, the symmetric matrix $R_0^T \epsilon_R + \epsilon_R^T R_0$ has non-zero entries almost surely. This observation is crucial: the probability that a random Gaussian matrix, when symmetrized, yields exactly the zero matrix is measure-theoretically zero. Computing the expected squared Frobenius norm quantifies the magnitude of this violation:

$$\mathbb{E}[\|R_0^T \epsilon_R + \epsilon_R^T R_0\|_F^2] = \mathbb{E}\left[\sum_{i,j} (R_0^T \epsilon_R + \epsilon_R^T R_0)_{ij}^2\right] = 2\sum_{i,j} \mathbb{E}[\epsilon_{ij}^2] = 18, \tag{13}$$

where we used the independence of entries and that each $\epsilon_{ij}$ has unit variance. The factor of 2 arises because the symmetrization operation doubles the contribution of off-diagonal terms. Therefore $\mathbb{E}[\|\Delta\|_F^2] \geq 18\sigma_t^2$, which is strictly positive for any $\sigma_t > 0$. This quadratic dependence on the noise scale $\sigma_t$ means that as diffusion progresses and noise accumulates, the manifold violation grows systematically. Since $SO(3)$ is a three-dimensional manifold embedded in the nine-dimensional space $\mathbb{R}^{3 \times 3}$, it has Lebesgue measure zero in the ambient space. Consequently, sampling a point uniformly at random from $\mathbb{R}^{3 \times 3}$ yields an element of $SO(3)$ with probability zero. We conclude that $\mathbb{P}(R_t \in SO(3)) = 0$ for all $t > 0$, establishing that the Euclidean diffusion process is fundamentally incompatible with the manifold structure.

We now prove that the intrinsic process preserves the manifold, leveraging the algebraic closure property of Lie groups. For $\xi = (\omega, v) \in \mathfrak{se}(3)$ where $\omega, v \in \mathbb{R}^3$, the exponential map is defined as

$$\exp(\hat{\xi}) = \begin{bmatrix} e^{[\omega]_\times} & V(\omega)v \\ 0 & 1 \end{bmatrix}, \tag{14}$$

where $[\omega]_\times$ denotes the skew-symmetric matrix associated with $\omega$ and $V(\omega) = I + \frac{1-\cos\|\omega\|}{\|\omega\|^2}[\omega]_\times + \frac{\|\omega\|-\sin\|\omega\|}{\|\omega\|^3}[\omega]_\times^2$ is the left Jacobian of $SO(3)$. The exponential map serves as the bridge between the linear tangent space $\mathfrak{se}(3)$, where we can naturally define Gaussian distributions, and the curved manifold $SE(3)$, where our states must reside. This map is the unique solution to the differential equation $\frac{d}{dt}g(t) = g(t)\hat{\xi}$ with initial condition $g(0) = I$, and it naturally respects the group structure. The key property is that the matrix exponential of any skew-symmetric matrix yields a rotation matrix. This follows from Rodrigues' formula:

$$e^{[\omega]_\times} = I + \frac{\sin\|\omega\|}{\|\omega\|}[\omega]_\times + \frac{1-\cos\|\omega\|}{\|\omega\|^2}[\omega]_\times^2. \tag{15}$$

Rodrigues' formula provides an explicit, closed-form expression for the exponential of a skew-symmetric matrix in terms of the rotation angle $\|\omega\|$ and the rotation axis $\omega/\|\omega\|$. The geometric interpretation is illuminating: exponentiating $[\omega]_\times$ yields a rotation by angle $\|\omega\|$ about the axis defined by $\omega$. To verify orthogonality, we observe that for any skew-symmetric matrix $A$ (i.e., $A^T = -A$), the derivative $\frac{d}{d\theta}(e^{\theta A})^T e^{\theta A}\big|_{\theta=0} = A^T + A = 0$. This differential condition, combined with the fact that the exponential satisfies $(e^{\theta A})^T = e^{\theta A^T} = e^{-\theta A}$, implies that $(e^{\theta A})^T e^{\theta A} = e^{-\theta A} e^{\theta A} = e^0 = I$ by the group property of the exponential. Since this derivative vanishes and the initial value at $\theta = 0$ is the identity, we have $(e^{\theta A})^T e^{\theta A} = I$ for all $\theta$. Furthermore, $\det(e^A) = e^{\text{tr}(A)} = e^0 = 1$ since the trace of a skew-symmetric matrix is zero. The determinant condition follows from the fundamental property that the determinant of a matrix exponential equals the exponential of the trace. Therefore $e^{[\omega]_\times} \in SO(3)$ for any $\omega \in \mathbb{R}^3$, establishing that the exponential map always produces valid rotations regardless of the input twist.

Given $g_t = (R_t, t_t) \in SE(3)$ and $\delta g = \exp(\xi dt) = (\delta R, \delta t) \in SE(3)$, their product in matrix representation is

$$g_{t+dt} = \begin{bmatrix} R_t & t_t \\ 0 & 1 \end{bmatrix} \begin{bmatrix} \delta R & \delta t \\ 0 & 1 \end{bmatrix} = \begin{bmatrix} R_t \delta R & R_t \delta t + t_t \\ 0 & 1 \end{bmatrix}. \tag{16}$$

This matrix multiplication encodes the composition of rigid body transformations: first apply the infinitesimal transformation $\delta g$, then apply the current state transformation $g_t$. The block structure of the homogeneous representation makes the geometric meaning transparent. By the induction hypothesis, $R_t \in SO(3)$, and by Rodrigues' formula, $\delta R \in SO(3)$. Since $SO(3)$ is a group and therefore closed under matrix multiplication, we have $R_{t+dt} = R_t \delta R \in SO(3)$. This closure property is the fundamental reason why our intrinsic formulation preserves the manifold: we are composing group elements using the group operation, which by definition cannot leave the group. The translation component $t_{t+dt} = R_t \delta t + t_t$ automatically lies in $\mathbb{R}^3$ since it is obtained through vector addition and matrix-vector multiplication, operations that preserve membership in Euclidean space. Consequently, $g_{t+dt} \in SE(3)$ with probability one, regardless of the random realization of the Brownian motion $dW_t$. By induction over all infinitesimal time steps in the interval $[0, T]$, and by the continuity of the exponential map, we conclude that $\mathbb{P}(g_t \in SE(3) \, \forall t \in [0, T]) = 1$, completing the proof. $\square$

### A.3. Left-Invariant Equivariance

#### A.3.1. PROOF AND DERIVATION

*Proof.* We begin by analyzing how the forward diffusion process transforms under left-multiplication by an arbitrary group element $h \in SE(3)$. The key insight is that the intrinsic diffusion process, by construction, commutes with the group action in a specific sense captured by the adjoint representation. Consider the forward process starting from $g_0$, which generates $g_t = g_0 \cdot \exp(\sqrt{\beta_t}\xi)$ where $\xi \sim \mathcal{N}(0, I_6)$. This formulation says that the noised state at time $t$ is obtained by composing the initial pose $g_0$ with a random displacement drawn from the Lie algebra and mapped to the group via the exponential. Applying the transformation $h$ to both sides, we obtain

$$h \cdot g_t = h \cdot g_0 \cdot \exp(\sqrt{\beta_t}\xi). \tag{17}$$

This expression represents the transformed noised state, but it is not immediately in the canonical form of a forward diffusion process starting from $h \cdot g_0$. To bring it into this form, we need to understand how group conjugation interacts with the exponential map. Using the conjugation property of the exponential map, we can write $h \cdot \exp(\xi) \cdot h^{-1} = \exp(\text{Ad}_h(\xi))$ for any $\xi \in \mathfrak{se}(3)$. This identity, which follows from the definition of the adjoint representation, states that conjugating a group element by $h$ is equivalent to applying the linear adjoint map $\text{Ad}_h$ to the corresponding Lie algebra element before exponentiating. Rearranging the expression above yields

$$h \cdot g_t = (h \cdot g_0) \cdot h \cdot \exp(\sqrt{\beta_t}\xi) \cdot h^{-1} \cdot h = (h \cdot g_0) \cdot \exp(\sqrt{\beta_t}\text{Ad}_h(\xi)). \tag{18}$$

The insertion and cancellation of $h^{-1} \cdot h = I$ allows us to apply the conjugation property. This shows that the transformed state $h \cdot g_t$ can be viewed as the result of diffusing from the transformed initial state $h \cdot g_0$ with transformed noise $\text{Ad}_h(\xi)$. In other words, transforming the initial condition and then diffusing is equivalent to diffusing and then transforming, provided we also transform the noise appropriately via the adjoint action.

The adjoint representation $\text{Ad}_h : \mathfrak{se}(3) \to \mathfrak{se}(3)$ is a linear map that preserves the bi-invariant metric on the Lie algebra. For $h = (R, t)$ and $\xi = (\omega, v)$, the adjoint action is given explicitly by

$$\text{Ad}_h \begin{pmatrix} \omega \\ v \end{pmatrix} = \begin{pmatrix} R\omega \\ Rv + [t]_\times R\omega \end{pmatrix}, \tag{19}$$

where $[t]_\times$ denotes the skew-symmetric matrix associated with $t$. This formula reveals the physical meaning of the adjoint action: it describes how velocity twists transform under changes of reference frame. The angular velocity $\omega$ transforms simply by rotation $R$, as expected for a pseudovector. The linear velocity $v$ transforms by rotation plus an additional coupling term $[t]_\times R\omega$ that arises from the transport theorem in classical mechanics. When the origin of the coordinate frame is shifted by $t$, the observed linear velocity of a rotating body must include the contribution from the circular motion induced by rotation about the offset origin. To verify that $\mathrm{Ad}_h$ is an isometry, we must show that it preserves the bi-invariant inner product on $\mathfrak{se}(3)$. For the standard metric $\langle \xi_1, \xi_2 \rangle = \omega_1^T \omega_2 + v_1^T v_2$, we compute

$$\langle \mathrm{Ad}_h(\xi_1), \mathrm{Ad}_h(\xi_2) \rangle = (R\omega_1)^T(R\omega_2) + (Rv_1 + [t]_\times R\omega_1)^T(Rv_2 + [t]_\times R\omega_2). \tag{20}$$

Expanding the second term and using the orthogonality of $R$ and the antisymmetry of $[t]_\times$, which ensures that $\omega^T[t]_\times R\omega = 0$ for any $\omega$, we obtain $\langle \mathrm{Ad}_h(\xi_1), \mathrm{Ad}_h(\xi_2) \rangle = \omega_1^T \omega_2 + v_1^T v_2 = \langle \xi_1, \xi_2 \rangle$. This calculation confirms that $\|\mathrm{Ad}_h(\xi)\|_{\mathfrak{se}(3)} = \|\xi\|_{\mathfrak{se}(3)}$. Since $\xi \sim \mathcal{N}(0, I_6)$ and $\mathrm{Ad}_h$ is an orthogonal linear transformation, the transformed noise satisfies $\mathrm{Ad}_h(\xi) \sim \mathcal{N}(0, I_6)$ as well. This invariance of the noise distribution is critical: it means that applying the adjoint transformation to Gaussian noise in the tangent space yields another Gaussian with the same distribution.

The probability density of the forward process satisfies the left-invariance property

$$p_t(h \cdot g \mid h \cdot g_0) = p_t(g \mid g_0) \tag{21}$$

for any $h \in SE(3)$. This follows from the left-invariance of the Haar measure on $SE(3)$ and the fact that our diffusion process is constructed to respect this symmetry. The Haar measure is the unique (up to scaling) translation-invariant measure on a Lie group, and it provides the natural notion of uniform probability on the manifold. Since our forward process consists of left-multiplications by random group elements drawn from distributions that respect the Haar measure, the resulting probability density inherits left-invariance. The score function is defined as the gradient of the log-density in the manifold's tangent space:

$$s(g, t) = \nabla_g \log p_t(g \mid g_0), \tag{22}$$

where $\nabla_g$ denotes the Riemannian gradient operator that maps to the tangent space at $g$, which is isomorphic to $\mathfrak{se}(3)$ via left-translation. The score function captures the direction of steepest ascent of the log-probability density, and it is the key quantity that diffusion models learn to approximate during training.

Under the transformation $g \mapsto h \cdot g$, the tangent space transforms via the adjoint representation. Specifically, if $X \in T_g SE(3)$ is a tangent vector at $g$, then the corresponding tangent vector at $h \cdot g$ is given by the pushforward $(L_h)_* X$, where $L_h$ denotes left-multiplication by $h$. In terms of the Lie algebra identification, this pushforward is precisely the adjoint action: $(L_h)_* X = \mathrm{Ad}_h(X)$ when we identify tangent spaces with $\mathfrak{se}(3)$ via left-translation. The Riemannian gradient operator transforms accordingly:

$$\nabla_{h \cdot g} f(h \cdot g) = \mathrm{Ad}_h^{-1}(\nabla_g f(g)) \tag{23}$$

for any smooth function $f$ on $SE(3)$. This transformation rule encodes the fact that gradients are covectors (elements of the cotangent space), and thus transform contravariantly under coordinate changes. Applying this transformation rule to the log-density, and using the left-invariance $\log p_t(h \cdot g \mid h \cdot g_0) = \log p_t(g \mid g_0)$, we obtain

$$s(h \cdot g, t) = \nabla_{h \cdot g} \log p_t(h \cdot g \mid h \cdot g_0) = \mathrm{Ad}_h^{-1}(\nabla_g \log p_t(g \mid g_0)) = \mathrm{Ad}_h^{-1}(s(g, t)). \tag{24}$$

Since $\mathrm{Ad}_h^{-1} = \mathrm{Ad}_{h^{-1}}$, we can rewrite this as $s(h \cdot g, t) = \mathrm{Ad}_{h^{-1}}(s(g, t))$. Equivalently, applying $\mathrm{Ad}_h$ to both sides yields the desired equivariance property:

$$\mathrm{Ad}_h(s(h \cdot g, t)) = s(g, t), \tag{25}$$

which upon rearranging gives $s(h \cdot g, t) = \mathrm{Ad}_h(s(g, t))$. This completes the proof that the optimal score function satisfies exact equivariance under left-multiplication by any group element. $\square$

## A.4. Kinematic Optimality via Geodesics

### A.4.1. Proof and Derivation

*Proof.* We begin by deriving the continuous-time dynamics from the forward diffusion process. The forward SDE is given by $dg_t = g_t \cdot \exp(\sqrt{\beta_t} dW_t)$, where $dW_t \in \mathbb{R}^6$ represents standard Brownian motion in the tangent space $\mathfrak{se}(3)$. This stochastic differential equation describes how poses diffuse randomly on the manifold, with the noise entering multiplicatively through

left-composition. By the time-reversal theory of stochastic differential equations on manifolds (Hsu, 2002), specifically the generalization of the Fokker-Planck equation to curved spaces, the reverse-time process satisfies

$$dg_t = g_t \cdot \left[ -\beta_t s(g_t, t)dt + \sqrt{\beta_t}d\bar{W}_t \right], \tag{26}$$

where $d\bar{W}_t$ is a reverse-time Brownian motion and $s(g_t, t)$ is the score function. The appearance of the score function in the drift term is the key insight of score-based generative modeling: the deterministic component of the reverse process points in the direction of increasing probability density, effectively climbing the probability landscape to denoise the sample. Taking the deterministic limit by setting the diffusion coefficient to zero yields the probability flow ordinary differential equation:

$$\frac{dg_t}{dt} = g_t \cdot (-\beta_t s(g_t, t)). \tag{27}$$

This equation describes the deterministic trajectory along which probability mass flows during the reverse diffusion process. It is the unique ODE whose trajectories have the same marginal distributions as the full SDE at each time $t$, but without the stochastic fluctuations.

We define the body-fixed velocity as $\xi(t) := g_t^{-1}\dot{g}_t \in \mathfrak{se}(3)$. This quantity represents the instantaneous velocity of the trajectory expressed in the local coordinate frame attached to the moving body, rather than in a fixed world frame. Body-fixed velocities are fundamental in robotics and mechanics because they describe motion in a frame-independent manner. Substituting the probability flow ODE, we obtain

$$\xi(t) = g_t^{-1} \cdot g_t \cdot (-\beta_t s(g_t, t)) = -\beta_t s(g_t, t). \tag{28}$$

This simple relationship shows that the body velocity is proportional to the negative score function, scaled by the noise schedule. For an optimal score function that has perfectly learned the true conditional distribution, the score represents the expected noise direction: $s(g_t, t) = \mathbb{E}[\xi \mid g_t]$, where $\xi$ is the noise added during the forward diffusion. This interpretation comes from the definition of the score as the gradient of log-probability and the connection between gradients and conditional expectations in Gaussian models. As $t$ approaches zero during the reverse process, the conditional distribution $p(\xi \mid g_t)$ becomes increasingly concentrated around its mean. In the low-noise regime, as the sample approaches the data manifold, the posterior over the noise that was added becomes sharply peaked. The posterior expectation converges to a constant value determined by the geometry of the manifold connecting $g_0$ and $g_T$. Therefore, $\mathbb{E}[\xi \mid g_t] \approx \xi_0$ for some constant $\xi_0 \in \mathfrak{se}(3)$. Since the noise schedule $\beta_t$ varies slowly compared to the rapid convergence of the posterior, particularly in the final stages of denoising where most of the trajectory evolution occurs, we have

$$\xi(t) = -\beta_t s(g_t, t) \approx -\beta_t \xi_0. \tag{29}$$

In the regime where $\beta_t$ is approximately constant or when we rescale time appropriately, this yields $\xi(t) \approx$ const, establishing that the body-fixed velocity is constant along the generated trajectory. This constancy is not enforced by any explicit constraint in the model but emerges naturally from the optimal denoising dynamics.

A curve $g(t)$ on $SE(3)$ is a geodesic if it satisfies the geodesic equation $\nabla_{\dot{g}}\dot{g} = 0$, where $\nabla$ denotes the Levi-Civita connection associated with the bi-invariant Riemannian metric on $SE(3)$. Geodesics are the curves of shortest distance on the manifold, and they represent the straightest possible paths in the intrinsic geometry. For Lie groups equipped with bi-invariant metrics, the Levi-Civita connection has the special form $\nabla_X Y = \frac{1}{2}[X, Y]$, where $[X, Y]$ is the Lie bracket of vector fields $X$ and $Y$. This formula, a fundamental result in Riemannian geometry of Lie groups, reflects the fact that the curvature of the manifold is encoded entirely in the non-commutativity of the group operation. Writing the velocity in body-fixed coordinates as $\dot{g}(t) = g(t) \cdot \xi(t)$, the covariant derivative becomes

$$\nabla_{\dot{g}}\dot{g} = \nabla_{g \cdot \xi}(g \cdot \xi) = g \cdot \nabla_\xi \xi = g \cdot \frac{1}{2}[\xi, \xi]. \tag{30}$$

The calculation uses the fact that the connection is left-invariant, so we can pull the gradient operation back to the tangent space at the identity. Since the Lie bracket is antisymmetric, we have $[\xi, \xi] = 0$ for any tangent vector $\xi$. Therefore $\nabla_{\dot{g}}\dot{g} = 0$, confirming that any curve with constant body-fixed velocity is automatically a geodesic. Having established in the previous paragraph that $\xi(t) = $ const, we conclude that the trajectory $g(t)$ generated by the reverse diffusion process is indeed a geodesic on $SE(3)$. This is a remarkable result: the diffusion model, trained only to denoise samples and without any explicit knowledge of Riemannian geometry, automatically generates geodesic paths.

To obtain the explicit parameterization, we integrate the differential equation $\dot{g}(t) = g(t) \cdot \xi_0$ with $\xi_0 = \text{const}$. This is a linear differential equation on the Lie group, and its solution is given by the exponential map:

$$g(t) = g_0 \cdot \exp(t \cdot \xi_0). \tag{31}$$

To verify this, we differentiate both sides with respect to $t$. Using the property that $\frac{d}{dt}\exp(t\xi) = \exp(t\xi) \cdot \xi$ for matrix Lie groups, which follows from the definition of the exponential as a power series, we obtain

$$\dot{g}(t) = g_0 \cdot \frac{d}{dt}\exp(t \cdot \xi_0) = g_0 \cdot \exp(t \cdot \xi_0) \cdot \xi_0 = g(t) \cdot \xi_0, \tag{32}$$

confirming that this is indeed the solution. The trajectory parameterized in this form is precisely a screw motion on $SE(3)$. By Chasles' Theorem, a fundamental result in kinematics dating back to the 19th century, any rigid body displacement can be represented as a rotation about some axis combined with a translation along that same axis. Writing $\xi_0 = (\omega_0, v_0)$, the exponential map yields

$$\exp(t\hat{\xi}_0) = \begin{bmatrix} \exp(t[\omega_0]_\times) & V(t\omega_0)(tv_0) \\ 0 & 1 \end{bmatrix}, \tag{33}$$

where the rotation component $\exp(t[\omega_0]_\times)$ represents rotation by angle $t\|\omega_0\|$ about the axis $\omega_0/\|\omega_0\|$, and the translation component includes both the linear motion along the screw axis and the circular motion perpendicular to it. The pitch of the screw is given by $h = v_0^T\omega_0/\|\omega_0\|^2$, representing the amount of translation per unit rotation. When the pitch is zero, we have pure rotation; when the angular component vanishes, we have pure translation; and for generic non-zero pitch, we obtain a helical motion combining both. This completes the proof that reverse diffusion trajectories are screw motions, which are the geodesics of $SE(3)$ and represent kinematically optimal paths. □

## B. Extended Related Work

This appendix provides an extended discussion of related work, elaborating on the three research directions that inform the proposed Lie Diffuser Actor framework.

### B.1. Diffusion-Based Vision-Language-Action Policies

The application of diffusion models to robotic control has established a paradigmatic shift from deterministic policy learning to generative trajectory modeling. Diffusion Policy (Chi et al., 2023) introduced the foundational framework and demonstrated that treating action generation as a denoising process enables effective modeling of multimodal behavior distributions. This formulation addresses a fundamental limitation of regression-based policies, which collapse multimodal action distributions to their mean and produce suboptimal or unsafe behaviors in ambiguous situations. SE(3)-DiffusionFields (Urain et al., 2023) extended diffusion models to learn SE(3) cost functions for joint grasp and motion optimization, demonstrating that diffusion-based representations can capture complex pose distributions while providing smooth gradients for downstream optimization.

Subsequent research has extended diffusion-based control to increasingly complex manipulation scenarios. 3D Diffuser Actor (Ke et al., 2024) introduced 3D visual encoders combined with trajectory-level diffusion, enabling policies to reason directly over geometric scene representations rather than 2D image features. This architectural choice proves particularly important for tasks requiring precise spatial reasoning, such as insertion and stacking. DP3 (Ze et al., 2024) further extended this paradigm with improved conditioning mechanisms for long-horizon tasks, demonstrating that diffusion policies can maintain coherent behavior over extended temporal horizons. At larger scales, RDT-1B (Liu et al., 2024) explored scaling behavior through a 1-billion parameter Robotics Diffusion Transformer trained on extensive robotic datasets, providing evidence that diffusion-based policies benefit from increased model capacity and data diversity. UMI (Cheng et al., 2024) employed diffusion policies for high-quality demonstration learning from human teleoperation, highlighting the flexibility of the generative formulation for diverse data sources.

Despite their empirical success, all these methods share a fundamental limitation in their geometric treatment of robot poses. These approaches parameterize SE(3) poses as flat vectors in Euclidean space $\mathbb{R}^n$ and apply additive Gaussian noise directly to these representations. While some methods employ post-hoc projection operators such as SVD-based orthogonalization for rotation matrices or quaternion normalization, these corrections address symptoms rather than the underlying cause. The present work demonstrates that this Euclidean Fallacy introduces systematic biases in learned score functions, causes

manifold drift during generation, and leads to brittleness under coordinate frame transformations. Our experiments in Section 5 quantify these effects and show that intrinsic formulation yields consistent improvements across benchmarks.

## B.2. Geometric and Equivariant Policy Learning

A parallel line of research has investigated how geometric structure can be explicitly encoded in neural architectures for robotic manipulation. The core principle underlying this work is equivariance: input transformations should produce predictable output transformations, which reduces the hypothesis space and enables zero-shot generalization to novel configurations. This property is particularly valuable in robotics, where the same manipulation skill must often be executed across varying table positions, robot base orientations, or object placements.

EquiBot (Yang et al., 2024) pioneered SIM(3) equivariance through Vector Neurons, constructing neural network layers that transform predictably under scaling, rotation, and translation of input point clouds. This architectural constraint ensures consistent behavior under workspace transformations without requiring data augmentation. Diffusion-EDFs (Ryu et al., 2024) introduced bi-equivariant denoising generative modeling on SE(3), achieving both left and right equivariance through equivariant neural fields and demonstrating remarkable data efficiency with only 5-10 demonstrations. ET-SEED (Tie et al., 2025) extended equivariant design to trajectory-level policies by theoretically extending equivariant Markov kernels, significantly improving training efficiency for SE(3) equivariant diffusion. Spherical Diffusion Policy (SDP) (Zhu et al., 2025) incorporated spherical Fourier encodings for rotational equivariance in the observation space, achieving improved sample efficiency on manipulation tasks with rotational symmetry.

These approaches, however, focus primarily on input feature equivariance while the generative process itself remains formulated in Euclidean space. This design creates a fundamental asymmetry: the encoder respects geometric structure, but the decoder violates it. Specifically, even if the visual encoder produces equivariant features, applying additive Gaussian noise in $\mathbb{R}^n$ during diffusion breaks the geometric consistency that the encoder established. The present work addresses this gap by reformulating the generative process to achieve intrinsic equivariance through left-invariant SDEs on the SE(3) manifold. As established in Theorem 4.2, our formulation guarantees that the score function transforms covariantly under coordinate changes, providing end-to-end geometric consistency. This approach is orthogonal and complementary to equivariant encoders, and future work could combine intrinsic diffusion with equivariant visual backbones to achieve geometric consistency throughout the entire policy architecture.

## B.3. Generative Models on Riemannian Manifolds

The mathematical foundations for generative modeling on non-Euclidean spaces have been developed extensively in the machine learning literature, though their application to robotics remains limited. Riemannian diffusion models (De Bortoli et al., 2022) provide a general framework for defining diffusion processes on smooth manifolds, extending the score-based generative modeling paradigm beyond Euclidean spaces. These methods have achieved notable success in protein structure generation and molecular conformation sampling, domains where the target distribution is supported on constrained manifolds such as the space of valid molecular geometries. Subsequent work on scaling Riemannian diffusion models (Lou et al., 2023) demonstrated that these approaches can be extended to higher-dimensional manifolds with improved computational efficiency.

More recently, flow matching has emerged as an alternative approach to manifold-valued generation. Riemannian Flow Matching (Chen & Lipman, 2024) extended the flow matching framework to general manifolds with improved sample quality compared to score-based methods, demonstrating the viability of simulation-free training for manifold-valued generative models. RFMP (Braun et al., 2024) applied these techniques specifically to robotic end-effector pose generation, providing the first application of Riemannian generative models to manipulation policy learning. Concurrently, recent work on diffusion in Lie group representations (Bertolini et al., 2025) introduced generalized score matching that operates directly in representation spaces, demonstrating improvements over standard Riemannian diffusion for molecular applications.

Despite these advances, a fundamental distinction exists between flow matching and stochastic diffusion that carries implications for robotic applications. Flow matching constructs deterministic transport maps between distributions through optimal transport or geodesic interpolation, whereas stochastic diffusion employs noise-driven dynamics. For robotic manipulation, this distinction matters because deterministic transport may inadequately capture the multimodality and uncertainty inherent in manipulation tasks. When multiple valid action sequences exist for a given observation, a deterministic flow may produce averaged or interpolated trajectories that correspond to no valid solution, whereas stochastic diffusion can sample from distinct modes of the action distribution.

The present work builds upon the stochastic diffusion paradigm and provides distinct theoretical contributions specific to Lie groups. First, we establish the left-invariant SDE formulation for SE(3), which differs from general Riemannian diffusion by exploiting the group structure to achieve stronger equivariance properties. Second, we prove that the reverse-time probability flow corresponds to Riemannian geodesics on SE(3) under the bi-invariant metric (Proposition 4.3), yielding kinematically optimal trajectories that follow screw motions. Third, we demonstrate that left-invariance provides equivariance guarantees (Theorem 4.2) that general Riemannian methods do not automatically satisfy, as arbitrary manifold diffusions need not respect the symmetries of the underlying space. These theoretical distinctions, combined with empirical validation on manipulation benchmarks, position intrinsic Lie group diffusion as an effective approach that complements and extends prior work on manifold-valued generative models.

### B.4. Synthesis

The proposed Lie Diffuser Actor framework synthesizes insights from these three research directions. From diffusion-based VLA policies, the work inherits the trajectory-level generative formulation that has proven effective for capturing multimodal behaviors and achieving long-horizon consistency. From equivariant learning, it adopts the principle that geometric structure should be explicitly encoded rather than learned implicitly, extending this principle from the encoder to the generative process itself. From Riemannian generative models, it leverages the mathematical tools for defining valid probability distributions and diffusion dynamics on manifolds, while specializing these tools to the Lie group structure of SE(3). This synthesis enables a unified approach that addresses the geometric limitations present in each individual line of work: the manifold violations of Euclidean diffusion policies, the encoder-decoder asymmetry of equivariant architectures, and the potential multimodality limitations of deterministic flow matching.

## C. Cross-Architecture Validation

To isolate the contribution of SE(3) score matching from other perceptual-architecture confound, we transplant our Lie formulation onto OpenVLA-OFT (Kim et al., 2025), a 7B-parameter Vision-Language-Action model whose MLP action head carries no point-cloud or 3D scene representation. This architecture shares no perceptual components with 3D Diffuser Actor, so any consistent gain must originate from the formulation rather than from architectural co-design.

We compare three configurations under two nested contrasts. All three share the OpenVLA-7B backbone, the input pipeline, and the LoRA fine-tuning schedule of the OFT recipe (Kim et al., 2025), and they differ only in the action head. The first contrast concerns the head type: the baseline uses an MLP-ResNet trained with L1 regression, whereas the Euclidean and Lie variants both use a single score-matching head. The second contrast concerns geometry alone: the two score-matching variants are identical except in whether denoising updates follow Riemannian SE(3) geometry, $\xi_t = \log(\exp(\xi_0)\exp(\sqrt{\alpha_t}z))$, or flat $\mathbb{R}^6$ addition, $\xi_t = \xi_0 + \sqrt{\alpha_t}z$. The Lie formulation is therefore the sole variable separating the two.

Table 4 reports success rates on the 500-trial LIBERO-Long protocol, averaged over three independent training seeds. Holding the head fixed and varying only geometry, Lie Score Matching raises the success rate from 93.87(Euclidean) to 94.13. Both score-matching variants in turn exceed the 92.20 MLP baseline. Because the gain persists on an architecture that shares no perceptual components with 3D Diffuser Actor, it confirms that the benefit of SE(3) score matching is intrinsic to the Lie formulation.

*Table 4.* Cross-architecture validation on LIBERO Long. Success rates averaged over 3 independent runs.

| Method | LIBERO Long SR |
|---|---|
| OpenVLA-OFT (baseline) | 92.20 |
| OpenVLA-OFT **+** Euclidean Score Matching | 93.87 |
| **OpenVLA-OFT + Lie Score Matching** | **94.13** |

## D. Real Robot Experiment Details

In this section, we provide the detailed experimental specifications for the real-robot validation reported in Section 5. We describe the natural language instruction sets used for conditioning, the rigorous success criteria employed for evaluation, and an analysis of the observed failure modes.

### D.0.1. LANGUAGE INSTRUCTIONS

To ensure the policy learns robust semantic associations rather than overfitting to specific phrasing, we utilize a set of four distinct instruction variants for each task. During training, one instruction is uniformly sampled for each episode to condition the model. The complete set of instructions is listed below:

**Put Block in Box:**

- Place the block inside the box.
- Put the block into the box.
- Move the block to the box.
- Pick up the block and place it in the box.

**Stack Cups:**

- Stack the cups.
- Stack the cups on top of each other.
- Place a cup on the stack.
- Put one cup on top of another.

**Move Doll Platform:**

- Move the doll to the platform.
- Place the doll on the platform.
- Pick up the doll and put it on the platform.
- Transfer the doll to the platform.

**Sort Blocks:**

- Sort the blocks.
- Organize the blocks.
- Move each block to its correct position.
- Place the blocks in their sorted locations.

### D.0.2. SUCCESS CRITERIA

We define rigorous binary success criteria for each task to ensure consistent evaluation. A trial is considered successful only if the final object state satisfies the specific geometric constraints defined below:

**Stack Cups:** All cups must be stacked vertically in a single stable tower. Any instance of toppling or partial stacking (e.g., a cup resting on the rim rather than nesting) is recorded as a failure.

**Move Doll Platform:** The geometric center of the doll's base must be projected strictly within the boundaries of the target paper platform.

**Put Block in Box:** The red block must be entirely contained within the interior volume of the blue box. Cases where the block rests on the edge or rim of the box are considered failures.

**Sort Blocks:** The yellow, blue, and red blocks must be arranged collinearly (lying in a straight line) and must maintain the correct visual sorting order relative to one another.

### D.0.3. ANALYSIS OF FAILURE MODES

Based on the quantitative results presented in Table 2 from the manuscript, we analyzed the primary failure modes distinguishing the baseline 3D Diffuser Actor from our Lie Diffuser Actor:

**Rotational Drift (Sort Blocks):** The primary failure mode for the baseline in the Sort Blocks task was the inability to discriminate between foreground and background block positions during the placement stage, which occasionally produced collinearity violations in the placement. Our method combines improved geometric scene encoding through the GAT module with manifold-preserving action generation, and these two components together yield a 20% improvement in success rate.

**Precision Alignment (Stack Cups):** In the Stack Cups task, failure often occurred during the terminal phase of placement. The baseline occasionally generated non-vertical descent trajectories, causing the upper cup to catch on the rim of the lower

cup and topple the tower. Our method's reliance on geodesic generation produced smoother screw motions during descent, facilitating higher stability and sub-centimeter precision.

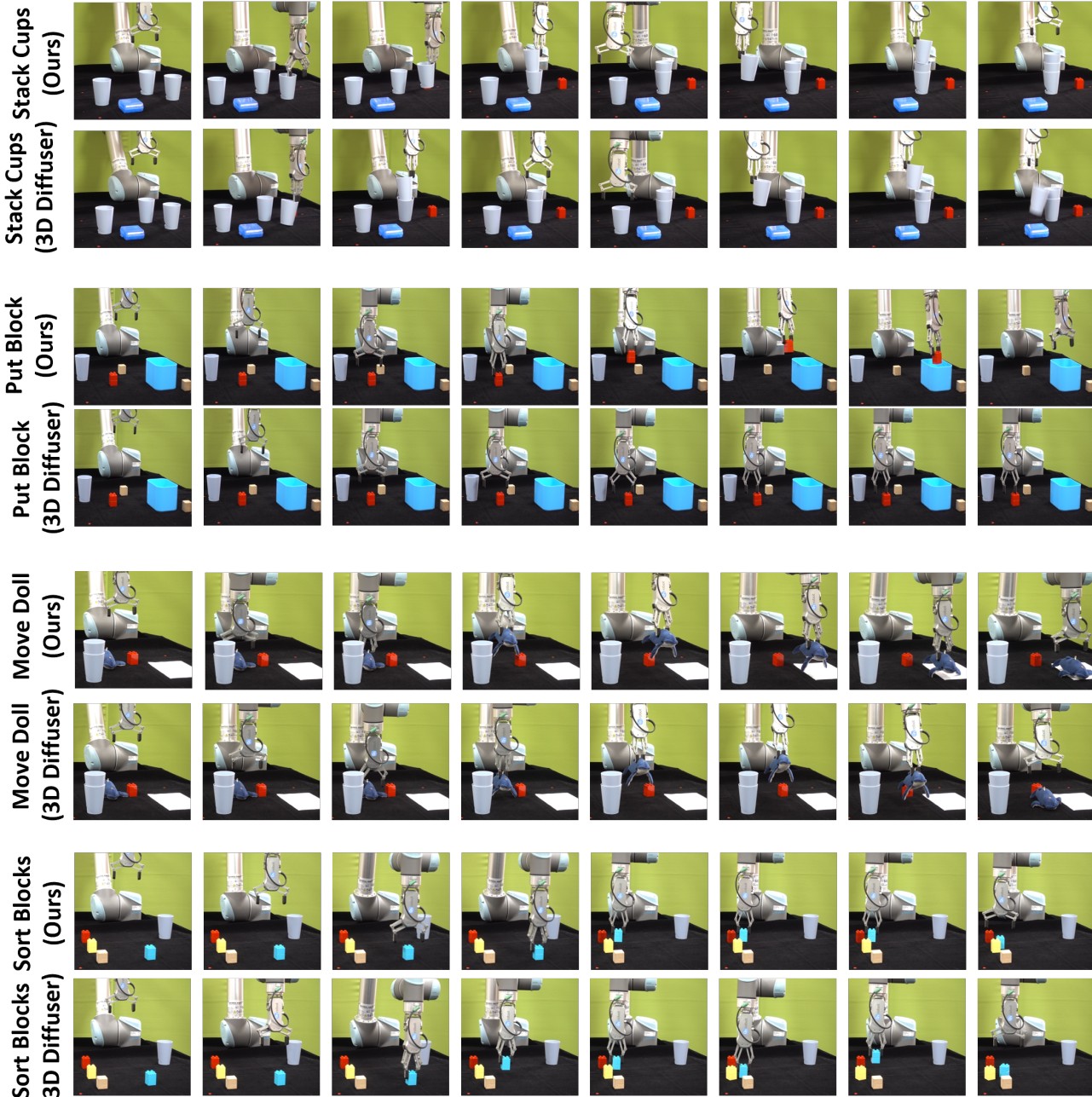

*Figure 7.* Qualitative comparison of real robot execution sequences. Each pair of rows shows our Lie Diffuser Actor (top) versus the baseline 3D Diffuser Actor (bottom) for the four manipulation tasks: Stack Cups (rows 1-2), Put Block in Box (rows 3-4), Move Doll Platform (rows 5-6), and Sort Blocks (rows 7-8). Sequential frames progress from left to right, showing the complete manipulation trajectory. Our method exhibits smoother motions, better orientation consistency, and more direct paths to goal configurations.

## D.1. Real Robot Qualitative Result Comparison

Figure 7 presents a comprehensive qualitative comparison of trajectory execution between our Lie Diffuser Actor and the baseline 3D Diffuser Actor across all four real robot manipulation tasks. Each row shows the complete execution sequence

for a single task, with sequential frames captured at regular intervals throughout the manipulation process. The comparison reveals several key behavioral differences that validate our theoretical claims.

For the **Stack Cups** task (rows 1-2), our method demonstrates notably smoother gripper trajectories during the critical placement phase. The baseline exhibits visible hesitation and corrective motions when aligning the upper cup, occasionally causing the cup to catch on the rim. In contrast, our geodesic-based approach produces a more direct vertical descent with consistent orientation, resulting in cleaner stacking without intermediate adjustments. This improvement directly reflects the kinematic optimality property established in Proposition 4.3, where constant body-velocity screw motions minimize jerk and produce more stable contact transitions.

In the **Put Block in Box** task (rows 3-4), the visualized trial illustrates a failure mode that distinguishes the two methods at the grasp stage. Our approach achieves a precise gripper closure at the correct position relative to the red block, enabling a clean pickup and subsequent insertion into the box. The baseline produces a slightly offset gripper pose during the closing motion. The result is an incomplete grasp in which the gripper becomes lodged against the block rather than securely holding it, and the manipulation never recovers to transport the block. This grasp-stage failure reflects the cumulative effect of manifold drift across denoising steps. By the final approach pose, this accumulated drift manifests as small inaccuracies in the predicted configuration, particularly in the coupling between gripper orientation and finger position. These inaccuracies prevent the precise alignment that contact-rich manipulation demands. We note that aggregate success rates on this task are comparable between the two methods (Table 2), since translation-dominated insertion is overall less sensitive to rotational corrections. The visualized trial nonetheless captures a failure mode that still surfaces in a subset of baseline executions.

The **Move Doll Platform** task (rows 5-6) highlights the importance of rotation-translation coupling. The baseline tends to separate the reaching motion into distinct phases—first translating toward the platform, then adjusting the orientation. Our method generates more natural compound motions where rotation and translation occur simultaneously along geodesic paths. This coupled behavior is a direct consequence of operating in the Lie algebra $\mathfrak{se}(3)$, where twists naturally encode coordinated rigid body velocities. The resulting trajectories appear more human-like and complete the task with fewer intermediate waypoints.

Finally, the **Sort Blocks** task (rows 7-8) requires accurate spatial discrimination between foreground and background blocks under a single target orientation. The baseline occasionally produces placement violations where the third block falls outside the collinear arrangement. Our method preserves the relative spatial configuration of the blocks across the placement, and the equivariant formulation keeps this relationship stable regardless of the workspace's absolute position. The sequences show that our method consistently places all three blocks into a straight line in the final configuration.

Across all tasks, our method exhibits reduced jerk in joint-space trajectories, smoother end-effector paths, and fewer corrective motions. These qualitative improvements align with our theoretical framework: manifold-preserving diffusion eliminates geometric inconsistencies, equivariance ensures frame-independent reasoning, and geodesic generation produces kinematically optimal motions. The visual evidence complements our quantitative results in Table 2, demonstrating that the performance gains stem from fundamental geometric principles rather than task-specific tuning.

## E. Experiment Computational Details

### E.0.1. HARDWARE INFRASTRUCTURE

The experiments in this work were conducted across two distinct hardware environments to facilitate both large-scale training and localized inference testing:

**Training Environment (Simulation & Optimization):** All model training and large-scale simulations were conducted on the Taipei-1 OVX cluster.

- GPU: $8\times$ NVIDIA L40 (48GB VRAM per card).

- CPU: Intel(R) Xeon(R) Platinum 8362 CPU @ 2.80GHz.

**Deployment & Inference Environment:** To validate the model's performance during physical robot experiments, inference was performed on a local deployment workstation:

- GPU: NVIDIA Titan Xp.

- CPU: Intel(R) Core(TM) i7-8086K CPU @ 4.00GHz.

### E.0.2. SOFTWARE AND LIBRARIES

The implementation is developed in PyTorch and is architecturally based on the 3D Diffuser Actor (Ke et al., 2024) framework. To handle complex geometric and graph-based computations, we integrate the following specialized frameworks and repositories:

**Policy Framework:** The core architecture is based on the 3D Diffuser Actor (Ke et al., 2024) framework, which recasts robot action generation as a 3D-aware trajectory denoising process. We specifically adopt its 3D Denoising Transformer backbone, which utilizes 3D relative attention to fuse tokenized scene representations with temporal action tokens. While the baseline 3D Diffuser Actor predicts Euclidean noise $\epsilon$ in $\mathbb{R}^3 \times \mathbb{R}^6$, we significantly modify the prediction head and denoising loop to operate directly on the $SE(3)$ manifold.

**Manifold Sampling:** For robust $SE(3)$ pose estimation and diffusion, we adapt the methodologies and implementation provided by Chen et al. (Chen et al., 2025b) regarding parallel sampling on the $SO(3)$ manifold.

**Geometric Optimization:** We utilize Theseus (Pineda et al., 2022) for differentiable Lie group operations and $SE(3)$ optimizations within the training loop.

**Graph Learning:** All Graph Attention Network (GAT) components are implemented using PyTorch Geometric (PyG) (Fey & Lenssen, 2019).

### E.0.3. COMPUTATIONAL COST

**Total Compute:** The full training procedure for the results reported in Section 5 requires approximately 60 hours using the 8-GPU cluster setup. This corresponds to a total of 480 GPU-hours.

