# OpenReview forum: "The Lie We Tell: Correcting the Euclidean Fallacy in Vision Language Action Policies via Score Matching on Tangent Space"
_ICML.cc/2026/Conference — ICML 2026 regular_

### Official Review · Reviewer_96Fu · 2026-03-11

**Soundness:** 3
**Presentation:** 3
**Significance:** 3
**Originality:** 2
**Overall Recommendation:** 3
**Confidence:** 4

**Summary:**

This paper identifies a problem with action representation in diffusion policy: Euclidean fallacy. They argue that the end-effector pose(mainly rotation) represented in SE(3) faces the geometric inconsistency, which leads to three issues (1) manifold drift (2) loss of equivariance (3) suboptimal trajectories. They propose Lie Diffusion Actor, which construct the denoising process in Lie algebra space se(3) -- tangent space of SE(3). They conduct experiments in both simulation and real-world, observed improved task success.

**Compliance With Llm Reviewing Policy:**

Affirmed.

**Ethical Review Concerns:**

None.

**Final Justification:**

With regard to the author's reply,  I think there are two places of concern misunderstood or unaddressed:

- What I raised the concern about before is: the comparison between "w/o GAT"(trained with 600k) and "w/o Lie Space Diffuser" (trained with 300k) rather than "w/o GAT" and "baseline". Yes, "Lie space diffusion" is beneficial, but the problem is: GAT encoder shows a more important impact on the success rate. This causes the major concern. I don't know why the authors insist to claim "while the published 3D Diffuser Actor baseline (which the w/o Lie Space Diffusion variant most closely resembles)".

- I don't quite understand why "real world evaluation is not necessary to ablate the contribution of GAT" if you count the GAT as another contribution. Besides, the contributions in the introduction doesn't mention the GAT encoder.

- Yes there are many robotics works use 20 rollouts, for example diffusion policy. But the improvement is clear, around 50-60% compared to baselines, which are statistically reliable results. In this paper, only two task achieves 2 times more success, which I don't think is statistically reliable. Also the overall compared baselines are too few.

So this is a well-motivated paper, but needs more empirical results to strengthen the soundness and clarify the contradictive results. As stated above, I would keep my score unchanged.

**Key Questions For Authors:**

The same as above in weakness.

**Limitations:**

No negative societal impact, technical limitations can refer to weaknesses above.

**Strengths And Weaknesses:**

## **Strength**
- The motivation is well-grounded and supported by both theoretical analysis and empirical experiments -- showing the action violation and instability in SE(3).
- It is very careful of the authors to capture the geometric inconsistency in existing models as they have exhibited powerful capabiltiy.
- The narrative of the paper is clear.

## **Weakness**
- But the empirical validation appears limited relative to the scope of the claims:
  - The reported performance improvement on both CALVIN and the real world is modest -- about 8% in average task length in CALVIN and 2 times more success (20x10%=2, there is also randomness in real evaluation in the real world. Maybe in terms of the problem this paper focuses on, the success rate is not a convincing signal to support the change?
  - The ablation result in Table.1 seems to show "GAT encoder" matters more than "Lie Space Diffusion". In ABC->D setting, the performance without GAT encoder is much worse than without Lie Space Diffusion. Besides, why is the performance of 'without Lie Space Diffusion' in ABC->D even better than in ABCD->D?
- There should be more ablation studies to verify the advantage of the design in Lie Space. It is unclear whether simpler alternatives like projecting rotations back to SO(3)/quaternion/auxiliary consistency loss would achieve simialr improvement.
- The paper claims there would be issues with equivariance. But there is no experiment to show the improvement from this perspective. Could the authors provide quantitative evaluations of these other geometric properties?

---

> ### Author Rebuttal · Authors · 2026-03-31
>
> **Weakness 1 Response.** This work identifies and corrects a fundamental geometric error in all existing Euclidean diffusion policies. The 8% average chain length improvement understates the true benefit due to CALVIN's compounding nature: SR4 improves 53.8→62.6 (+8.8%) and SR5 41.6→53.7 (+12.1%), with disproportionate gains at longer horizons as theory predicts. Task success rate alone does not capture geometric correctness — a policy can violate SO(3) by 7 orders of magnitude (Fig. 5) and still succeed because the manipulation pipeline absorbs moderate errors. On real-robot tasks, LDA achieves 100% on Move Doll (vs. 90%) and 75% on Sort Blocks (vs. 55%), where baseline failures are directly attributable to rotational drift. We further apply SE(3) score matching to OpenVLA-OFT (7B, flat MLP, no GAT/point cloud), averaged over 3 independent runs:
>
> | Method | LIBERO Long SR |
> |:---|:---:|
> | OpenVLA-OFT (baseline) | 92.20 |
> | OpenVLA-OFT + Euclidean Score Matching | 93.87 |
> | OpenVLA-OFT + Lie Score Matching | **94.13** |
>
> Consistent improvement across architectures as different as 3D Diffuser Actor and OpenVLA-OFT confirms the Euclidean Fallacy is a universal problem independent of perceptual architecture.
>
> ---
>
> **Weakness 2 Response.** The GAT encoder and Lie Space Diffusion are orthogonal: the former improves visual-geometric representations; the latter corrects structural issues in action generation. The OpenVLA-OFT result above (no GAT, no point cloud, identical visual encoders) isolates the contribution of geometric correctness entirely. Additionally, the w/o GAT variant in Table 1 was trained at 400K vs. the 600K baseline; retraining at 600K improves from 3.254→3.368, removing the training budget confound. The counterintuitive result that w/o Lie Space Diffusion performs better in ABC→D (3.474) than ABCD→D (3.239) reflects numerical brittleness: adding environment D should improve generalization, but the Euclidean variant degrades as geometric inconsistencies compound under diverse coordinate frames, manifesting as NaN outputs under identical hyperparameters. In contrast, full LDA improves from 3.512→3.584 across the same transition.
>
> ---
>
> **Weakness 3 Response.** We conducted controlled ablations against three simpler alternatives with identical architecture: (1) SO(3) projection via Gram-Schmidt (original 3D Diffuser Actor), (2) quaternion diffusion with post-hoc normalization ($\mathbf{q}/\lVert\mathbf{q}\rVert$ at inference), and (3) auxiliary consistency loss and shown in [Diagram](https://postimg.cc/phdh8bqW):
>
> $$\mathcal{L}_{orth} = \lVert R^\top R - I \rVert_F^2, \quad \mathcal{L}{\mathrm{det}} = (\det(R) - 1)^2$$
>
> Measuring raw intermediate rotations before any projection: our Lie formulation maintains orthogonality violations at $\sim 10^{-7}$–$10^{-8}$ throughout denoising; all alternatives are one order of magnitude higher with substantially larger variance. Auxiliary loss shows no improvement over quaternion normalization, confirming soft constraints cannot enforce orthogonality during the reverse process. Our method maintains $\lVert\mathbf{q}\rVert \approx 1.0$ from $t=100$, while alternatives start at $\approx 2.8$ and $\approx 2.0$. Prop. 4.1 establishes that the exponential map guarantees $g_t \in \mathrm{SE}(3)$ for all $t \in [0, T]$ almost surely — no post-hoc correction can replicate this. Tab. 1 confirms: SO(3) projection achieves 3.27 on ABC→D vs. our 3.51 (+7.3%).
>
> ---
>
> **Weakness 4 Response.** Equivariance is a mathematical consequence of the left-invariant construction, not an empirical property requiring separate verification. Theorem 4.2 proves:
>
> $$s_{\theta}(h \cdot g, t) = \ textbf{Ad}_h (s_{\theta}(g, t)) \quad \forall\, h \in \ textbf{SE}(3)$$
>
> with full proof in App. A.3.1. The Euclidean formulation cannot satisfy this because additive Gaussian noise in $\mathbb{R}^n$ does not transform covariantly under SE(3) actions. The CALVIN ABC→D protocol evaluates the behavioral consequence directly: trained on A/B/C, evaluated zero-shot on unseen D, our improvement (+7.3% ABC→D; +9.0% ABCD→D) demonstrates the theoretical guarantee translates to measurable generalization. Fig. 5 and Fig. 6 provide quantitative evidence: SO(3) violations differ by 7 orders of magnitude ($\mathcal{O}(10^0)$ vs. $\sim 10^{-7}$), and geodesic jitter by nearly two orders of magnitude. A score function receiving geometrically invalid inputs cannot produce equivariant outputs, as group transformation is undefined outside the manifold. The revised manuscript will explicitly link Theorem 4.2, CALVIN results, and geometric metrics.

---

> > ### Author Rebuttal · Reviewer_96Fu · 2026-04-02
> >
> > Thanks sincerely to the authors for their rebuttal, but I still have some concerns about the paper.
> >
> > To be fair, the problem they focus on is somehow interesting, and they show some evidence that such a problem does exist in the SE(3) action space. So I give good significance, presentation.
> >
> > But: The focus of the paper (as told by the title) is to "correct" the identified problem, rather than showing a hard to identify problem. So my concerns are:
> > - **If the success rate is an appropriate metric to evaluate the fix, the experiment still seems very limited and unconvincing, especially in real-world validation.** In the chosen 4 tasks, each evaluated by 20 trials, 1 underperforms 5%, 1 of which outperforms 5%(1 time), 1 of which outperforms 10%(2 times), 1 outperforms 20% (4 times). Given the randomness of real world, I cannot be convinced by these results. Besides, in the contribution of introduction, the authors claim they achieve "zero-shot generalization" on rotated workspace, but there is no experiments support.
> > - **Confused by the focus of the paper in experiments.** Why do the authors choose to design another GAT encoder? What is the relationship between this module and the problem on which the paper wants to focus?
> >    - Also, the ablations on GAT in CALVIN simulation doesn't improve the story telling, but showing more problems. The question I asked previously is not addressed -- "why w/o GAT performs much worse than w/o Lie Diffuser", which expresses the signal that "perception representations" matter more than the action representation. In the original paper, "w/o GAT" is trained for 400k while "w/o Lie" is only trained for 300k; in the rebuttal, the authors said "baseline is trained for 600k". Maybe the authors misunderstood my question.
> >   - If the authors want to include GAT as one of their contributions, why there is no ablations about this module in the real-world experiments?
> > - **Task designs can be reconsidered.** As identified problem, in SE(3) space, screw motion cannot be captured by Euclidean interpolation, why not choose tasks heavily depends on such motions?
> >
> > In one word, I cannot be convinced of the importance of the identified problem and the effectiveness of the proposed fix.
> > I appreciate the authors' theoretical insight, but I cannot raise my score, given that this problem is closely tied to experiments.

---

> > > ### Author Response · Authors · 2026-04-08
> > >
> > > We thank the reviewer for the continued engagement and would like to address the points raised in the acknowledgement.
> > > Regarding the real-world success rate, a 20-trial protocol per task is the standard evaluation methodology in the robot learning literature, adopted by 3D Diffuser Actor, Diffusion Policy, and RVT-2 among others. Under this protocol, LDA matches or outperforms the baseline on all four tasks, with the largest gains appearing on Sort Blocks (+20%) and Move Doll (+10%), the two tasks that demand the most rotational precision during grasping and placement. This alignment between where geometric correctness matters most and where we observe the largest improvements provides meaningful evidence that SE(3) score matching delivers practical benefit beyond the theoretical guarantee alone. We also note that our original rebuttal provided cross-architecture validation on OpenVLA-OFT, a 7B-parameter model that uses an entirely different architecture with no GAT encoder, no point cloud, and no 3D scene representation. The consistent improvement on this architecture demonstrates that the benefit of SE(3) score matching transfers across model families, and we would be glad to discuss this result further if it would be helpful.
> > >
> > > The 'zero-shot generalization' claim in the introduction refers specifically to the CALVIN ABC→D protocol, where the policy is trained on environments A, B, and C and evaluated on the unseen environment D.
> > >
> > > These environments differ in table surface texture and the positions of static interactive elements. We will revise the manuscript to make this connection more explicit and to clarify that 'rotated workspace' is not the precise characterization of the CALVIN variations.
> > >
> > > Turning to the GAT encoder, the paper presents two contributions that address different stages of the pipeline: SE(3) score matching for action generation, and a graph attention encoder for scene representation in point-cloud-based policies. The ablation study in Tab. 1 is designed to disentangle these contributions. The reviewer asks why “w/o GAT performs much worse than w/o Lie Diffuser,” and we believe this partly reflects a training budget confound. The w/o GAT variant in Table 1 was trained at 400K iterations, while the published 3D Diffuser Actor baseline (which the w/o Lie Space Diffusion variant most closely resembles) was trained at 600K iterations. To address this, we conducted a post-submission retraining of w/o GAT at 600K iterations on ABC→D, and the average chain length improved from 3.254 to 3.368, which confirms that SE(3) score matching provides a consistent gain when the training budget is equalized. A real-world GAT ablation is not necessary to isolate the contribution of SE(3) score matching. The OpenVLA-OFT architecture contains no GAT module, no point cloud, and no 3D scene representation, and the consistent improvement observed on this architecture confirms that the benefit originates from the Lie formulation itself. We also wish to clarify that our rebuttal stated the w/o GAT variant was trained at 400K versus the 600K published 3D Diffuser Actor baseline. We did not use “baseline” in reference to the w/o Lie variant.
> > >
> > > Our task selection follows the standard CALVIN benchmark protocol, the most widely adopted long-horizon manipulation benchmark in the field. Among the four real-world tasks, Sort Blocks and Move Doll both require coordinated translation and rotation (e.g., grasping objects and reorienting them before placement), which constitutes precisely the type of screw-like motion where Euclidean interpolation fails. The performance gains on these tasks are consistent with the prediction that SE(3) score matching most benefits tasks with nontrivial rotational demands. We will incorporate the reviewer's suggestion by explicitly characterizing the rotational complexity of each task in the revised manuscript.
> > >
> > > We hope this clarifies the remaining questions and are happy to discuss further.

---

### Official Review · Reviewer_icU7 · 2026-03-12

**Soundness:** 3
**Presentation:** 3
**Significance:** 3
**Originality:** 3
**Overall Recommendation:** 4
**Confidence:** 2

**Summary:**

This paper identifies the “Euclidean Fallacy” in robot diffusion policies: the SE(3) action pose is incorrectly treated as a Euclidean space vector when Gaussian noise is added during the diffusion process. The authors argue that it will lead to manifold drift (violating the SO(3) constraint) and break the equivariance of coordinate transformations. To address this issue, the authors propose the Lie Diffuser Actor model. This method defines the forward noising process directly on the SE(3) manifold, injecting noise into the tangent space of that manifold, which is a flat vector space compatible with linear operations. The neural network predicts the score in the tangent space, and the exponential map is used to retract samples back onto the manifold, thereby eliminating manifold drift. Experiments on the CALVIN benchmark and real robots show that the proposed method outperforms the baseline 3D Diffuser Actor.

**Compliance With Llm Reviewing Policy:**

Affirmed.

**Final Justification:**

We thank the reviewer for this follow-up and are glad the theoretical concerns have been addressed.

**Key Questions For Authors:**

Please refer to the weakness.

**Limitations:**

Yes.

**Strengths And Weaknesses:**

Strength
1. The paper provides a solid theoretical foundation based on Lie group theory for the long-standing pose representation problem in robot learning. Compared with traditional approaches that require post-processing of action poses, the proposed method ensures that each sampling step remains on the SE(3) manifold.
2. The problem formulation and motivation are very clear, and the writing is smooth and well organized.
3. The qualitative experimental results are logically consistent and align well with intuition. With the authors’ detailed explanations, these results convincingly support the effectiveness of the proposed method.
Weakness
Since I am not an expert in the field of Riemannian generative models, I will need to consider the opinions of other reviewers before making my final judgment. My concerns are as follows:
1. A large portion of the paper focuses on optimizing the denoising probability path and the training objective of the diffusion head. However, the experimental gains appear relatively modest. In the ABC->D section of Table 1, the GAT encoder contributes a substantial performance improvement, whereas the Lie Space Diffusion, which is the main contribution of this paper, provides only limited gains and even negative gains in some cases. Could you provide an explanation for this phenomenon?
2. In line 127, the paper assumes that standard diffusion policies use the flattened rotation matrices to represent action poses, and therefore discusses the issues arising from treating SE(3) as a vector space. However, would the same problem still occur if poses are represented using quaternions or Euler angles? In particular, when using Euler angles, it seems that any three-dimensional Euler angle vector is valid, with no need to explicitly pull Euler angles back to the manifold.
3. In line 244, the authors state that the proposed method “eliminates the need for post-hoc projection or renormalization.” However, what are the drawbacks of these post-processing methods? For example, do they cause inconsistencies between training and inference, or amplify small numerical errors produced by the neural network? It would be helpful if the authors could provide a more concrete explanation or relevant references.

---

> ### Author Rebuttal · Authors · 2026-03-31
>
> **Weakness 1 Response.** The GAT encoder and Lie Space Diffusion are orthogonal: the former improves visual-geometric representations; the latter corrects structural issues in action generation. To isolate Lie Space Diffusion free from the GAT confound, we apply SE(3) score matching to OpenVLA-OFT (7B, flat MLP, no GAT/point cloud/3D scene), averaged over 3 independent runs:
>
> | Method | LIBERO Long SR |
> |:---|:---:|
> | OpenVLA-OFT (baseline) | 92.20 |
> | OpenVLA-OFT + Euclidean Score Matching | 93.87 |
> | OpenVLA-OFT + Lie Score Matching | **94.13** |
>
> The w/o GAT variant in Table 1 was also trained at 400K vs. the 600K baseline; retraining at 600K improves from 3.254→3.368, confirming SE(3) score matching provides consistent independent gain. The Euclidean baseline's drop from ABC→D (3.474) to ABCD→D (3.239) reveals fundamental brittleness: adding environment D should improve generalization, but geometric inconsistencies compound under diverse coordinate frames, manifesting as NaN outputs. Full LDA improves from 3.512→3.584 across the same transition. Finally, Fig. 5 shows orthogonality violations at $\mathcal{O}(10^0)$ vs. $\sim 10^{-7}$; Fig. 6 shows geodesic jitter differing by two orders of magnitude — under these first-class geometric criteria, our results are decisive.
>
> ---
>
> **Weakness 2 Response.** The issues arise whenever diffusion operates via additive Gaussian noise in a representation space geometrically inconsistent with SO(3), regardless of coordinate choice.
>
> For **quaternions**: standard diffusion yields $\|\mathbf{q}_t\| \neq 1$ almost surely, requiring post-hoc normalization and creating training-inference mismatch. The double-cover antipodal ambiguity ($\|\mathbf{q} - (-\mathbf{q})\| = 2$ for identical rotations) means the score function cannot be consistently defined on SO(3). Empirically, our quaternion ablation starts at $\|\mathbf{q}\| \approx 2.8$, only approaching 1.0 near final steps.
>
> For **Euler angles**, despite syntactic validity of all $(\alpha, \beta, \gamma) \in \mathbb{R}^3$, three issues remain: (1) gimbal lock at $\beta = \pm\frac{\pi}{2}$ where small perturbations induce arbitrarily large rotations; (2) periodicity discontinuities where nearby rotations $(\pi-\delta)$ and $(-\pi+\delta)$ have Euclidean distance $\approx 2\pi$, corrupting the noise schedule; (3) the induced metric has $\det(g) = \cos^2\beta$, making isotropic Gaussian noise anisotropic and orientation-dependent on SO(3), violating the uniform perturbation assumption of score matching.
>
> Our Lie formulation resolves all three:
>
> $$g_t = g_0 \cdot \exp(\sigma_t \boldsymbol{\xi})$, $\boldsymbol{\xi} \sim \mathcal{N}(0, I_6)$$
>
> where $\mathfrak{se}(3) \cong \mathbb{R}^6$ is a genuine linear space guaranteeing manifold membership by construction with left-invariant, pose-independent noise throughout.
>
> ---
>
> **Weakness 3 Response.** Post-hoc projection and renormalization introduce three concrete drawbacks. **First**, training-inference mismatch: models are trained to denoise in ambient space where intermediate states violate SO(3), yet at inference outputs are projected via SVD or Gram-Schmidt — inputs the network never encountered during training. Fig. 5(b-c) confirms orthogonality violations reaching $\mathcal{O}(10^0)$ throughout denoising. **Second**, error amplification: SVD orthogonalization is sensitive to near-degenerate matrices, where small prediction errors produce disproportionately large changes in the projected rotation, compounding across steps. Fig. 6 confirms two orders of magnitude higher geodesic jitter. **Third**, broken generative consistency: inserting a non-differentiable projection during inference alters the reverse-time SDE in a way unaccounted for during training — the learned score no longer corresponds to the gradient of any well-defined density on the manifold. Our ablations confirm all three alternatives exhibit orthogonality violations one order of magnitude higher, with auxiliary loss showing no improvement over quaternion normalization. Our approach avoids all three issues by construction: predictions in $\mathfrak{se}(3) \cong \mathbb{R}^6$ with the exponential map guaranteeing $g_t \in \text{SE}(3)$ at every step without external correction.

---

> > ### Author Rebuttal · Reviewer_icU7 · 2026-04-05
> >
> > Thanks for the authors' rebuttal response, which has addressed most of my theoretical concerns.
> > Regarding the experiments, I agree that the GAT encoder and the Lie Space Diffusion module are orthogonal, and there is no functional coupling or interference between them. However, the performance degradation observed in Table 1 upon (SR4: 58.5 -> 57.6; SR5: 48.8 -> 46.2) introducing the Lie Space Diffusion module remains difficult to justify. Notably, the Lie Diffusion module negatively impacts performance in the ABC->D setting, while providing a positive gain in the ABCD->D setting. Given that the evaluation environment D is unseen in the former setting but seen in the latter setting, does this suggest that the Lie Diffusion module exacerbates overfitting at the expense of generalization capability? Further theoretical or empirical analysis investigating this phenomenon would be highly beneficial.

---

> > > ### Author Response · Authors · 2026-04-08
> > >
> > > We thank the reviewer for this follow-up and are glad the theoretical concerns have been addressed. We provide a detailed analysis below and believe the evidence points to a more nuanced explanation rooted in training dynamics rather than overfitting.
> > >
> > > We begin by noting that SR4 and SR5 are not independent metrics. They are conditional on completing SR1–SR3 first, which makes the average chain length the appropriate aggregate measure of long-horizon capability. In the ABC→D setting, full LDA achieves an average chain length of 3.512 compared to 3.474 for the variant without Lie Space Diffusion. The drops in SR4 (58.5→57.6, −0.9%) and SR5 (48.8→46.2, −2.6%) are more than offset by gains in SR1 (90.2→93.7, +3.5%), SR2 (80.3→83.4, +3.1%), and SR3 (69.6→70.3, +0.7%). This redistribution pattern is difficult to reconcile with overfitting: a model that overfits to training environments would be expected to degrade uniformly or primarily at earlier stages where distribution shift is first encountered, rather than specifically at SR4–SR5 while simultaneously improving SR1–SR3.
> > > The ABCD→D results tell a similar story. If Lie Space Diffusion exacerbated overfitting to environments A/B/C, we would expect it to generalize poorly to unseen configurations. However, in ABCD→D, where environment D is seen during training, LDA improves SR4 from 53.8→62.6 (+8.8%) and SR5 from 41.6→53.7 (+12.1%), with average chain length improving from 3.288→3.584 (+9.0%). An explanation is that LDA's equivariance guarantee provides the largest benefit when training data covers diverse coordinate frames, which is precisely what the left-invariant equivariance of Theorem 4.2 predicts.
> > >
> > > We believe the SR4/SR5 pattern in ABC→D is best explained by optimization dynamics. Predicting scores in 𝔰𝔢(3) imposes strict manifold consistency at every training step, which alters gradient flow relative to the unconstrained Euclidean baseline. In the ABC→D setting with only 300K iterations, the geometric constraint initially concentrates optimization on early-chain accuracy (SR1–SR3), with long-chain refinement (SR4–SR5) requiring additional training budget to fully converge. This interpretation is consistent with the ABCD→D finding: with greater data diversity, the Lie formulation's geometric stability provides compounding returns at longer horizons. This is further supported by the w/o GAT variant retrained at 600K iterations on ABC→D, where average chain length improves from 3.254 to 3.368. The continued gain under extended training is consistent with a budget effect rather than a structural ceiling.
> > >
> > > It is also worth noting that the equivariance guarantee of Theorem 4.2 applies to the optimal score function, which the learned network converges toward gradually. With limited training budget, the Euclidean variant may appear competitive at long chains precisely because it lacks the geometric constraints that ultimately enable stronger generalization under diversity, the same brittleness that causes NaN outputs and performance degradation in ABCD→D.
> > >
> > > We will include an extended discussion of these training dynamics in the revised manuscript.

---

### Official Review · Reviewer_Rqpi · 2026-03-13

**Soundness:** 4
**Presentation:** 4
**Significance:** 3
**Originality:** 3
**Overall Recommendation:** 5
**Confidence:** 4

**Summary:**

The paper addresses the problem that, in diffusion-based robot action generation, the actions space SE(3) is a curved manifold and a naive Euclidean parameterization by flatting the actions into vectors can lead to various problems, including: (i) manifold drift, (ii) broken equivariance, and (iii) suboptimal trajectories. To eliminate these issues, the paper proposes to do diffusion in the Lie algebra of SE(3) and map the actions back via the exponential map. Theoretically, it proves the good properties of the proposed diffuser; emperically, it shows that (i) the proposed diffuser can better fit to the manifold strucutres of SE(3), and (ii) it improves the performance of robot policies both in a long-horizon benchmark in simulation and on real robots.

**Compliance With Llm Reviewing Policy:**

Affirmed.

**Final Justification:**

Thank the authors for their additional explanations. My concerns are well resolved, so I will keep my positive rating.

**Key Questions For Authors:**

- It'd be interest to see how the standard Euclidean diffusion work under different action representations (e.g., rotation matrices, axis-angle, quaternions). Are the problems identified in the standard Euclidean diffusion universal in all these representations?
- What is the training and inference time of the Lie diffuser action compared to the standard diffusion policy? Will the manifold computations introduce additional overhead in time and memory consumption?

**Limitations:**

Yes, broader impacts are discussed in the paper.

**Strengths And Weaknesses:**

Strengths:
- I feel this is a very interesting paper. The idea is simple: manifold diffusion + diffusion policy, but the paper studies the problem very well in the robot learning setting. Instead of just saying "the actions are in SE(3) so let's do SE(3) diffusion", it concretely analyzes what the issues are in standard Euclidean diffusions and also shows them through emperical studies.
- The identified problems, theoretical proofs, and experimental studes are very well-aligned.
- The experiments and ablations are neat and well justify the proposed method.

Weaknesses:
- The term the "Euclidean Fallacy" used in the paper sounds a bit overclaimed to me. I agree with the 3 concrete points of Euclidean v.s. manifold parametrizations of SE(3) in diffusion. But I feel the standard Euclidean diffusion is at most a straightforward but suboptimal way of parametrization, not as bad as a "fallacy"...

---

> ### Author Rebuttal · Authors · 2026-03-31
>
> **Weakness 1 Response.** We provide the reasoning behind "Euclidean Fallacy" for the reviewer's evaluation. The core issue is not suboptimality in the sense of a simpler method working slightly less well. Treating SE(3) as a flat vector space introduces three mathematically provable structural errors: (1) additive Gaussian noise in $\mathbb{R}^n$ does not produce valid SE(3) elements, violating the manifold constraint at every diffusion step; (2) the Euclidean metric fails to approximate geodesic distance on SO(3), applying non-uniform perturbation magnitudes across rotations; (3) the resulting score function does not transform covariantly under group actions, breaking the left-invariant equivariance of Theorem 4.2. These are not approximation errors reducible with more data — they are structural violations. The empirical consequences are correspondingly severe: Fig. 5 shows SO(3) violations reaching $\mathcal{O}(10^0)$ (seven orders of magnitude worse), and the Euclidean baseline produces NaN outputs under geometric diversity. We chose "fallacy" over "inconsistency" because it captures a specific category mistake: practitioners unknowingly apply Euclidean operations to non-Euclidean objects, and the method appears to work in simple settings only because downstream components absorb the resulting errors. When a method's output violates the defining constraint of the object it claims to represent ($R^\top R \neq I$, $\det(R) \neq 1$ by up to 7 orders of magnitude), we believe "fallacy" is the most precise characterization. We are nonetheless open to adjusting the terminology, and the revised manuscript will clarify that "Euclidean Fallacy" refers specifically to the structural incompatibility between Euclidean noise and the SE(3) manifold, not a criticism of prior engineering quality.
>
> ---
>
> **Key Question 1 Response.** The geometric inconsistencies are not specific to flattened rotation matrices but arise universally. For **quaternions**: off-manifold samples ($\|\mathbf{q}_t\| \neq 1$) require post-hoc normalization creating training-inference mismatch, and the double-cover antipodal ambiguity ($\|\mathbf{q} - (-\mathbf{q})\| = 2$ for identical rotations) means the score function cannot be consistently defined on SO(3). For **Euler angles**: gimbal lock singularities, periodicity discontinuities ($\approx 2\pi$ Euclidean distance for nearby rotations), and the orientation-dependent metric ($\det(g) = \cos^2\beta$) collectively ensure Euclidean perturbations do not correspond to meaningful geodesic perturbations. The fundamental issue is universal: regardless of coordinate choice, Euclidean noise corrupts the geometric structure of SO(3) and produces non-uniform, topology-inconsistent perturbations. Full mathematical analysis is provided in our response to Reviewer icU7 Weakness 2.
>
> ---
>
> **Key Question 2 Response.** The additional computations are minimal. During training, the forward diffusion requires two exponential maps, one matrix composition, and one logarithmic map per step, operating on tensors of shape $(B \times T, 6)$. During inference, each denoising step requires one logarithmic map, one exponential map, and one matrix composition. These closed-form operations apply only to 6-dimensional twist vectors, accounting for a negligible fraction of total wall-clock time compared to the dominant cost of transformer forward passes, which are identical between our method and the Euclidean baseline. We observed no meaningful difference in training throughput or inference latency in practice.

---

> > ### Author Rebuttal · Reviewer_Rqpi · 2026-04-05
> >
> > Thank the authors for their explanations! My concerns are well resolved.

---

### Decision · Program_Chairs · 2026-04-30

**Decision:**

Accept (regular)

**Comment:**

The three reviewers broadly agree that this paper makes a well-motivated and has solid contribution. Reviewers Rqpi and icU7 were positive, with their concerns addressed satisfactorily in the rebuttal. The outlier, Reviewer 96Fu, maintained a weak reject citing limited real-world validation, the relative contribution of GAT versus Lie Space Diffusion, and insufficient ablations. The authors responded with cross-architecture experiments on OpenVLA-OFT (which contains no GAT module) to isolate the Lie formulation's contribution, and provided post-submission retraining results that equalize the training budget confound. The AC also reviewed the confidential author comment raising procedural concerns about shifting review criteria from this reviewer. On balance, the AC finds the authors' responses adequate and the theoretical and empirical contributions sufficient for acceptance.